# Combining acoustic survey and citizen science data yields enhanced species distribution models for tropical rainforest birds

Reid Rumelt[1]*, Carla Mere Roncal[2], Arianna Basto[2], Zuzana Buřivalová[3], Christopher Searcy[1]

**1** Department of Biology, University of Miami, Coral Gables, Florida, United States of America, **2** New Venture Fund, Washington, DC, United States of America, **3** The Nelson Institute for Environmental Studies and the Department of Forest and Wildlife Ecology, University of Wisconsin-Madison, Madison, Wisconsin, United States of America

* rumeltr@miami.edu

## Abstract

A key goal in ecology is to develop effective ways to understand species' distributions in order to facilitate both their study and conservation. Many species distribution modeling analyses have been performed using either structured survey data or unstructured citizen science data; these two pools of data have tradeoffs in terms of data density, spatiotemporal coverage, and accuracy. Recent studies have shown that combining structured and unstructured survey data can improve the accuracy of species distribution models for birds, but most of this work has focused on north temperate bird species, using bird atlas data that are less available in the Tropics. Here, we adapted a data pooling approach from the literature on north temperate bird biology to create distribution models for a selection of secretive suboscine bird species that occur in a highly diverse region of the southwestern Amazon. Our approach combined automated acoustic monitoring detections and eBird citizen science data available for the region as well as a high resolution land cover dataset of the region's key ecological gradients. The pooled models outperformed models constructed solely with eBird data for predicting fine grain species responses to habitat gradients in intact forest, but also retained information from the citizen science dataset about species occurrence patterns in non-vegetated areas away from intact forest, including those subject to human disturbance. We present this hybrid approach as a flexible and repeatable means to produce inferences that would not easily be achievable using a single data source, and provide recommendations for other researchers seeking to replicate these methods in Amazonia as well as in other tropical regions.

**Data availability statement:** The minimal dataset required to replicate all study findings ("data-raw.zip") as well as all code used to produce the study findings are available here at https://doi.org/10.5281/zenodo.15276321. In addition, an annotated dataset of WAV files representing the BirdNET detections for all five target species, along with sound selection files compatible with the Raven Pro analysis program, is available at https://doi.org/10.5281/zenodo.15711887.

**Funding:** This work was funded by National Science Foundation Graduate Research Fellowship Grant 2023305561 to RBR (http://www.nsf.gov), a Lewis and Clark Research Grant from the American Philosophical Society (no grant number issued, https://www.amphil-soc.org/grants/lewis-and-clark-fund-explora-tion-and-field-research), a Franzen Fellowship grant to RBR (no grant number issued, https://www.amazonconservation.org/tag/jona-than-franzen-fellowship/), and Prince Albert II of Monaco Foundation grant 3386 to ZB. The funders had no role in study design, data collection and analysis, decision to publish, or preparation of the manuscript.

**Competing interests:** The authors have declared that no competing interests exist.

## Introduction

Quantifying the environmental factors that influence organismal occurrence patterns has helped address key research questions in 21st century biology [1–3]. These include describing life history strategies [4,5], estimating the relative contributions of ecological partitioning and competitive interactions in community structuring [6], and predicting threats due to anthropogenic change [7]. A key tool in this research is species distribution modeling, which relates information on species occurrence (presence, presence-absence, and/or abundance) to environmental predictors [8–11]. Due to longstanding historical biases in biological research, most of this work has focused on north temperate species [12]. There are still key areas in which tropical ecosystems lack high quality organismal data relative to temperate ecosystems, par-ticularly with regard to spatial survey bias. However, the steady accumulation of data over time and the development of novel types of surveying techniques means that applying species distribution modeling to vastly more speciose tropical communities is more tenable now than it has ever been [13,14].

Acoustic monitoring is a key emerging technique for surveying tropical rainforest birds, as it requires limited person-hours in potentially difficult-to-access field sites, is more effective at detecting species living in heavily vegetated understory than camera traps, and can be used to survey for long periods at a time [14–16]. The ability of automated recording devices to collect large volumes of data at specific sites also makes this survey technique a critical tool in areas that are likely to expe-rience significant anthropogenic disturbance in the near future and for which limited time exists to gather data [14–16]. As acoustic surveys involve careful *a priori* study design, they can be used to collect data that selectively targets local-scale habitat gradients thought to be important to the study species and have known and control-lable amounts of spatial and temporal bias [17–19]. By repeatedly sampling survey sites, researchers can also use these data in occupancy models to jointly predict occurrence and detection processes [20]. However, while automated recording units (ARUs) are capable of recording continuously for many days or weeks given suffi-cient battery and storage capacity, the need to purchase several units given the small area of coverage of each (~150 m radius around the recorder [21]), and the field labor required for servicing means that monitoring projects using ARUs are typically focused on high density of sampling sites rather than wide spatial coverage [22]. An important research priority is therefore to find effective ways to combine acoustic survey data with other datasets to make inferences that are generalizable to larger spatial areas.

Citizen science datasets have shown great potential to fill spatial gaps in ecologi-cal data [23–26] as they generally have very high data densities, are free to use, and cover broad spatial scales. However, they also typically contain lower evenness of observer quality and higher levels of spatiotemporal survey bias relative to expert-lead surveys [27,28]. A variety of research has examined how to mitigate sources of bias in citizen science data [17,19,29–32]. The largest citizen science project for birds is the eBird project [33–35], with >100 million complete eBird checklists (observations

which include the list of species observed and a variety of associated effort data) submitted by citizen science observers around the world [36]. Species distribution modeling methods that use eBird data, typically paired with 500 m spatial resolution Moderate Resolution Imaging Spectroradiometer (MODIS) landcover data to describe important factors mediating occurrence, have been used by ornithologists to generate inferences about bird biology for over a decade [30,37–41].

Several recent publications have demonstrated that combining eBird data with small amounts of structured survey data collected from the same geographic area and time period (referred to as "data pooling") is a simple yet powerful way to increase the predictive power of eBird species distribution models (SDMs) [17,42,43]. As structured surveys can be designed to provide more even coverage of relevant habitat gradients than is typically found in citizen science datasets, data pooling has increased the capacity of eBird SDMs to identify important habitat associations of threatened bird species [42]. Data pooling is likely to be especially useful in areas where eBird data densities are low and unevenly distributed [42,44]. This could help address the long-standing asymmetry in the application of SDM in the Tropics relative to temperate ecosystems. However, prior attempts to apply integrated modeling approaches to eBird data have focused on North American birds and achieved these results using structured survey data from The Nature Conservancy and the Breeding Bird Atlas [42,43]; at present, similar large bodies of survey data are largely unavailable in the global Tropics. As a consequence, an important research goal in tropical ornithology is to show that integrated modeling with eBird data can generalize well to other forms of structured survey data, including acoustic monitoring data, that can serve as a replacement for surveys conducted by skilled technicians in regions where the latter is limited or nonexistent.

The Amazon rainforest is one of the most speciose regions for birds globally [45,46]. Major drivers of diversity in this region include rivers, other habitat discontinuities [47–49], and interspecific competitive interactions [50,51], but also habitat heterogeneity and long-term ecological stability [52–55]. Using novel modeling frameworks to assess how habitat gradients mediate bird occurrence patterns could help answer important questions about community assembly processes and habitat partitioning within clades of Amazonian birds that up to this point have been difficult to answer due to data and survey effort limitations.

Here, we apply a framework that integrates data from acoustic surveys with eBird observations to model the occurrence of several species of secretive interior forest suboscine songbirds in a highly diverse region of primary forest in southwestern Amazonia. This is the first time an integrated modeling framework combining acoustic monitoring and citizen science data has been applied to Amazonian birds. We aim to evaluate whether this data pooling approach, combined with a higher resolution land cover dataset of the key ecological gradients in this region, is capable of producing inferences that are not possible with models using solely acoustic or solely eBird data. In addition, we test whether data pooling improves the ability of models to capture local-scale habitat preferences in tropical forest, as has previously been shown to be true for temperate systems [42]. We test the applicability of this methodology using interior rainforest birds whose habitat preferences make them better candidates for surveying with automated acoustic monitoring than with traditional field survey methods. Finally, we offer suggestions and caveats to applying this methodology to other study systems in the Neotropics as well as other regions across the global Tropics.

## Methods

### Survey site and study species

All acoustic survey data were collected at Los Amigos Biological Station and the adjacent conservation concession of the same name, which contains approximately 145,000 hectares of lowland primary rainforest in the province of Madre de Dios in southeastern Peru. Recording was conducted on the station's existing transect grid, covering an area of approximately 4000 hectares, which was previously used for camera trap studies [56,57]. Habitat heterogeneity in this region is primarily the result of successional gradients associated with increasing distance from the channels of large rivers [58], and is coincident with high avian alpha diversity (~600 species [36]). The two high-level forest types present in the southwestern Amazon are floodplain forest (flooded for at least part of the year) and *terra firme* (never flooded). Differences in

time since disturbance, soil moisture, and edge effects are responsible for creating distinct understory vegetation characteristics in these two habitats as well as the ecotone between them [58,59]. Members of several avian clades regularly segregate spatially by forest type, suggesting that high habitat and species diversity metrics are related [60–62]. eBird survey data were collected from a much broader area within Madre de Dios, some of which is covered by intact forest but elsewhere contains open lands and areas of anthropogenic land use. Madre de Dios currently experiences significant threats from deforestation and fire, and the area bordering the *Ruta Interoceánica*, a newly-constructed east-west road connecting Madre de Dios and the Peruvian Andean provinces with the Brazilian state of Acre, is a major deforestation hotspot within Peru [63]. Madre de Dios has also become a hotspot for illegal artisanal gold mining in recent years; direct removal of vegetation near rivers and oxbow lakes and large inputs of mercury into the environment have yielded lasting and widespread land use change in this region, even deep into otherwise intact forest areas [15,64,65].

We focused our analyses on a set of five suboscine songbirds in Furnariida, an extremely speciose clade in Amazonian forests: *Formicarius analis, F. colma, Myrmothera campanisona*, *Akletos goeldii,* and *Oneillornis salvini* [36,66] (Table 1), all of which are well-represented in acoustic monitoring data available for the station. These species are all terrestrial or semi-terrestrial, cryptically-colored, and have strong affinities for intact forest with heavy understory vegetation [67], making them important indicators of forest quality [68,69]. At the same time, these species are quite distinct from one another in terms of their utilization of the region's key ecological gradients, and these differences are well-documented in the literature (Table 1). We used these known differences to our advantage in this analysis by assessing the extent to which they were recapitulated in our candidate models.

## Acoustic data collection

Acoustic data used for model training were collected during three recording seasons. The first two seasons represent existing data from a pilot study and subsequent full monitoring project for tinamous (*Tinamidae*), respectively [76]. For these projects, 10 Cornell Lab of Ornithology Swift autonomous recording units (ARUs) were cycled across 34 survey points covering the terra firme-floodplain gradient (Fig 1) from late July to early October 2019 (dry season in southeastern Peru) and then again at the same sites from early January to late February 2020 (wet season) [56]. Recording durations were 2–23 days per site during the pilot study and 21 days per site during the wet season. The third season ran from mid-May to mid-September 2024, focusing on 19 new recording sites that were located along the same trail system. These new sites were chosen to provide better spatial resolution within the ecotone between terra firme and floodplain (Fig 1). From May to mid-August, 4 Frontier Labs BAR-LT recorders were deployed for three 30-day long deployments at 4 sites. Starting in mid-August and continuing through the end of the season, 6 Cornell Lab of Ornithology SwiftOne ARUs and 9 Frontier Labs BAR-LT recorders were deployed for an additional single 21-day long deployment at the remaining 15 sites. ARUs were placed no closer than 150 m apart, as this is believed to be the cone of acoustic detectability in dense tropical forest [21]. Recorders were placed ~1 m above the ground with the microphone facing towards the ground, which is optimal for recording understory (most activity ≤ 2 m above ground) and terrestrial birds. Additional details of recording site selection, placement, and recording parameters for the three seasons are available in citation [14] as well as in S1 File.

**Table 1. List of species used in analyses.**

| Common Name | Scientific Name | Family | Habitat Preferences | Citations |
|---|---|---|---|---|
| **Black-faced Antthrush** | *Formicarius analis* | Formicariidae | Floodplain and transition forest | [70,71] |
| **Rufous-capped Anthrush** | *Formicarius colma* | Formicariidae | Terra firme | [70,71] |
| **Thrush-like Antpitta** | *Myrmothera campanisona* | Grallariidae | Terra firme and transition forest | [72,73] |
| **Goeldi's Antbird** | *Akletos goeldii* | Thamnophilidae | Floodplain forest with heliconias and bamboo | [72,74] |
| **White-throated Antbird** | *Oneillornis salvini* | Thamnophilidae | Terra firme and transition forest, obligate ant-follower | [72,75] |

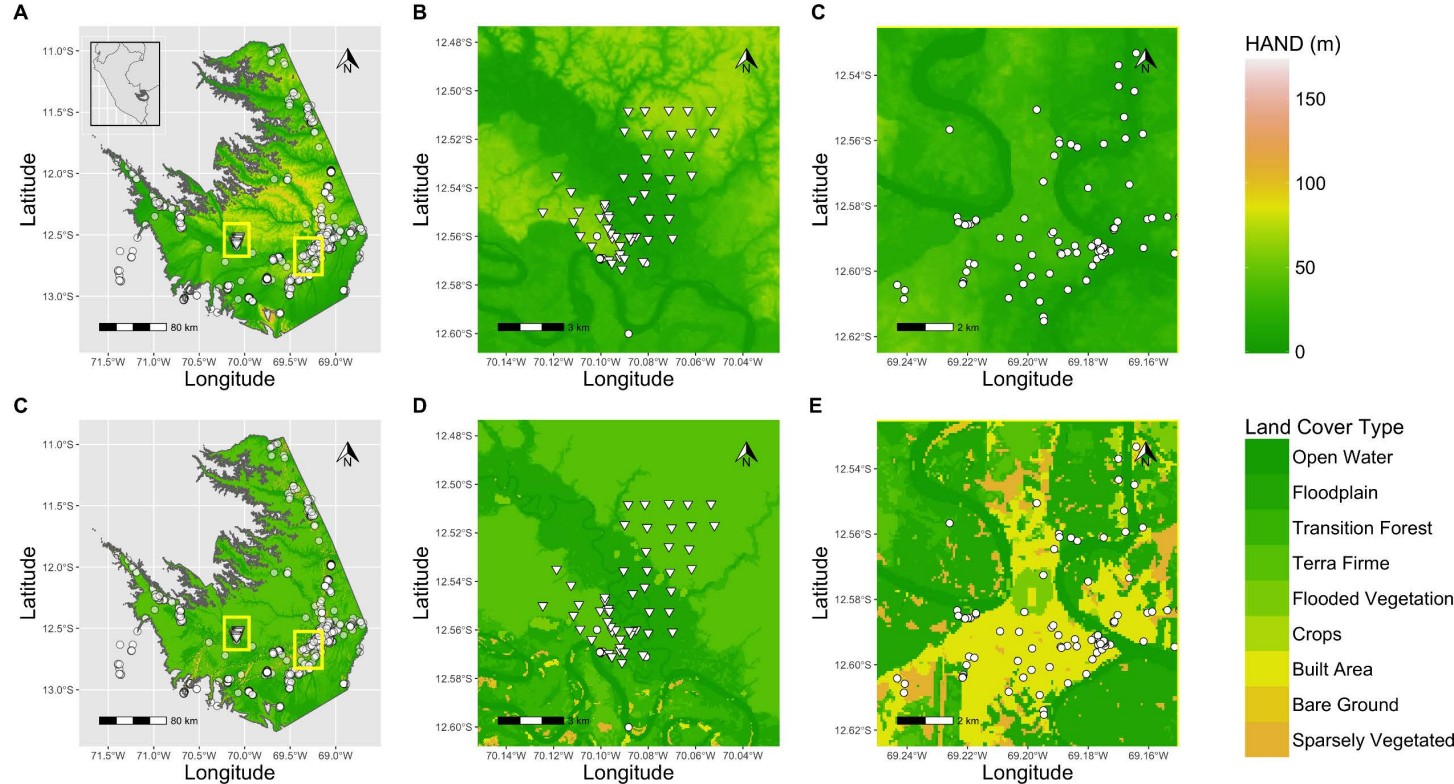

**Fig 1. Spatial distribution of datasets.** eBird checklists (circles) and recording sites (triangles) within Madre de Dios (A,D), at Los Amigos Biological Station (B,D), and near Puerto Maldonado and nearby agricultural areas (C,F) with respect to the height above nearest drainage (HAND) gradient (A-C) and land use types (D-F). Recording sites were selected to span the HAND gradient (B), while eBird sites occur in a diversity of habitat types. The western yellow box in (A,C) denotes extent of (B,D), while the eastern yellow box denotes extent of (C,F). Study region is outlined in inset map of Peru in (A).

## Processing of acoustic data

Survey audio was captured as a series of WAV files (see S1 File for details on file length) spanning each day's recording period. Processing was performed using BirdNET-Analyzer version 2.4, which is pre-trained on a list of ~6000 bird species that includes our five target species and much of the avifauna of Amazonian Peru [77]. The classifier examines 3-second moving windows spanning the source audio and returns possible detections with associated detection probabilities (see S1 Table for full classification parameters). A characteristic of all probability-based moving window classifiers is the need to set a minimum threshold for accepted detections. To accomplish this, we performed a semi-supervised calculation of thresholds for each species based on literature best practices for BirdNET confidence score interpretation [78]. We took a sample of the positive detections for each species spanning the full confidence score range of the classifier output (0.3–1); in order to more evenly cover the spatial gradient at Los Amigos, we stratified this sample such that up to 30 detections were chosen from each 10 m interval of elevation across acoustic sampling sites (200–293 m). Species with low average BirdNET classifier performance may have many more low confidence score detections than high confidence score detections, thus reducing sample size in the range of scores where false positives transition to true positives [78]. To account for this, we added a second independent sample of up to 30 detections at a range of higher scores (0.8–1) from each 10 m elevational band. We labeled these detections as true or false using the annotation function of the Raven Pro software suite [79], and fit three candidate models to these labels and their associated logit-scaled confidence scores: i) a fixed-intercept null model, ii) a generalized linear model with logit score as the predictor and label as the response, and iii)

a logistic generalized additive model (family "binomial" with REML optimization) with logit score (linear term) as a predictor and label as the response, but with the addition of site elevation (smooth term, cubic regression spline) as an environmental covariate to account for the possibility that classifier performance might vary across the habitat gradient due to the presence of similar-sounding organisms across certain sections of the gradient. Candidate models were evaluated using the Akaike Information Criterion (AICc); for all five species analyzed, the best performing model was the logistic GAM (S2 Table). The threshold for accepting detections was calculated for each species as the lowest (e.g., most permissive) threshold that kept the false positive rate (*fpr*) as evaluated on each species' annotated sample set below 5% [78]. Acoustic detections for each species were assigned probability scores using the best fit model and filtered using the chosen threshold. We performed this thresholding process separately for the 2019–2020 seasons and the 2024 season to better account for differences in recording quality and microphone SNR between years, and possible increases in ambient noise at some sites in 2024 due to increased mining activity along the nearby river. In addition, we temporally filtered the 2024 detections so that they matched the diel recording schedule of the 2020 season to ensure greater consistency in survey effort (S1 File). After all other sampling evenness filtering was performed, we collapsed sequences of vocal events into at most one detection per species per hour of recording time [62,80]; we refer to these as "independent detections."

To facilitate joining acoustic and eBird data for use in model creation, we formatted the acoustic detection CSV file so that it contained the same columns as the eBird dataset download. Using the '*auk*' package in R [81], we compared the detection set to a database that contained a list of all one-hour recording periods across the field seasons with associated site-date information. Each hour-long recording period received a 1 for each of the target species that was present and 0 for each of the target species that was absent. We hereafter refer to this type of presence-absence dataset taking into account all sampling events as "zero-filled," as this is the term used in the literature on eBird SDM [82].

### eBird data and landcover covariates

For model training, occurrence data for the five target species through September 2024 were acquired from the eBird data portal [35]. Data were pre-processed using the '*auk*' package in R [81] to select checklists containing at least one of our five target species that were submitted in Madre de Dios between 2010–2023 using semi-structured protocols (Stationary or Traveling). To further increase sampling effort evenness within the dataset, we filtered this dataset to contain only complete checklists (those where the observer specifies that they reported all species they were able to identify) with survey duration less than five hours, effort distance less than 1 km (for traveling checklists), and number of observers less than 5. While these filtering constraints are broadly consistent with what is typically used in the literature on eBird SDM [82], our shorter distance cutoff (1 km as opposed to 5 km) accounts for the fact that ecological gradients in the Tropics tend to be shorter than in the temperate zone [83]. Following the literature on eBird SDM [82], we recoded all checklists with "X's" for our target species as NAs, which removed them from the dataset. We repeated the previously-described zero-filling process by joining the eBird checklists to the complete database of eBird checklists from the same geographic area, time period, and effort parameters such that each checklist received a 1 if the target species was present and 0 if the target species was absent. Although the set of absences in this dataset could include events where the species was undetected despite actually being present, eBird data are generally treated this way in the SDM literature as they are considered to be a sufficiently good approximation of actual presence-absence processes [84]. In addition, we consider our target species to be very easy to detect by eBirders: as well as having loud, simple vocalizations, they are also among the most common bird species in Madre de Dios (reported more often than 98.0% of all other species in Madre de Dios for *F. analis*, 87.5% for *A. goeldii*, 75.4% for *O. salvini*, 74.2% for *F. colma*, and 64.0% for *M. campanisona*). In any case, the measures of occurrence used in this analysis either incorporate detectability indirectly through covariates (encounter rate) or directly through repeated site samples (occupancy), so we felt this issue is of limited enough scope to consider our data to be presence-absence.

Prior analyses using eBird data for species distribution modeling [82] have typically used the Moderate Resolution Imaging Spectroradiometer (MODIS) land-use/land-cover dataset [85] to predict species habitat affinities, with elevation

as an auxiliary continuous predictor variable. The MODIS land cover dataset lumps most Amazonian forest types into an "evergreen broadleaf forest" class, and has a resolution (500 m) that is too coarse to resolve the short ecological gradients present in the region [56,86,87]. As a result, we felt it necessary to develop a custom predictor dataset that is resolved to a finer spatial scale and contains more nuanced land cover classes that are relevant for analyses in the Amazon. In order to better capture the natural gradient that exists between terra firme and floodplain forest types in lowland Madre de Dios, we chose to base our analysis mainly on a height above nearest drainage (HAND) surface [59] calculated from the SRTM+ 30 m Digital Elevation Model (DEM) [88] and a linear feature map of the main river channels in the region [89]. We chose to use HAND instead of elevation since, as the primary habitat gradient in this region is maintained by fluvial processes, HAND was likely a more direct proxy for flooding intensity than elevation and accounts for the increase in mean riverbed elevation as one moves west across our study region towards the Andean foothills [56]. We based our categorical land cover predictor set on a Sentinel-2 10 m Land Use/Land Cover Time Series dataset that contains 11 land cover classes (Fig 1) [90]. This land cover layer includes several non-forest land categories (such as bare ground, crops, and built areas); while none have non-forest values at the acoustic points (preliminary data analysis indicates that all of the acoustic points are ≥ 1 km from any non-forest land cover classes), we believed they would be crucial for measuring absences for eBird sites in non-forest areas. Ground-truthing of the Madre de Dios corridor also indicates that dredge tills and temporary habitations on river beaches subject to alluvial gold mining [15] are typically (correctly) classified as "built," and we felt including these categories might also help our models capture the effects of this form of disturbance on species presence at eBird sites along rivers. We enlarged the size of class four of the Sentinel-2 dataset ("Flooded Vegetation") by superimposing two land cover classes ("occasionally open water" and "always inundated") from a 30 m resolution synthetic aperture radar dataset of forest inundation [91] to create a more all-encompassing "flooded vegetation" land cover class. We also included a canopy height predictor based on data from the Global Ecosystem Dynamics Investigation (GEDI) LIDAR project and the Sentinel-2 dataset [92]. All raster values were resampled (nearest neighbor for categorical data, bicubic for continuous data) to 30 m for consistency.

Species distribution modeling is sensitive to spatial scale, and inferences become less robust with increasing distance in spatial and environmental space from the region encompassed by model training data [93]. To ensure that predictions would not be inferred at sites far away from the original sampling locations, we constructed our model prediction surface by creating buffers around each eBird and acoustic monitoring point that we merged into a single polygon and clipped to the borders of Madre de Dios department. We used 35 km as the buffer width for this step as this was the smallest distance which ensured that only a single polygon was left after merging. In addition, we clipped this region with a polygon containing all areas of Madre de Dios with elevations ≤ 350m. All occurrence data used in model training were filtered to exclude points outside this prediction surface, leaving us with data collected at sites with similar elevations and climates to the area where acoustic monitoring was conducted. All GIS layer manipulation was performed using QGIS version 3.26.0 [94].

## Modeling framework

Our modeling approach follows that used in earlier studies of eBird data pooling [42,43,82] unless otherwise noted. Although eBird data can be used to model abundance [82], we chose not to construct abundance models in this analysis because our acoustic data are not well suited to capturing this metric. We created three general "classes" of models for this analysis: acoustic only models, eBird-only models, and models containing data from both sources, the latter of which were created by row-joining the two datasets and adding a column labeling each row as an acoustic or eBird observation. All model processing was performed in R version 4.2.0 [95].

## Encounter rate

Encounter rate (sometimes also called "encounter probability") is a commonly used metric in the eBird SDM literature [42,84]. Although it is not capable of isolating the factors that determine site occupancy (z), it is nonetheless a useful

approximation of this quantity as it is approximately equal to the product of the site occurrence probability (ψ) and detectability (p) processes from occupancy modeling and uses observation date, time, duration, distance covered, and number of observers as covariates in order to account for between-site variability in detection [82,96]. Model outputs are read as the probability that an average observer will encounter the species given one hour of survey effort during the time of day at which the target species is most frequently recorded [82]. To produce encounter rate models for our species and evaluate the benefit of including structured survey data from the acoustic dataset, we used a bootstrapping approach to fit models on random samples of audio, eBird, and pooled data (n = 100 per model type per species). In each bootstrap iteration, the eBird dataset for our target species was spatiotemporally subsampled by overlaying the sample area with a 600 m x 600 m grid and sampling one observation per week within each cell. This particular grid size was chosen as it is the smallest number that ensures that eBird sample points are no more clustered than the acoustic survey points (mean nearest neighbor distance = 599.34 m). Positive and negative detections were sampled separately to reduce class bias [18]. Spatiotemporal subsampling was not performed on the acoustic survey data because our strategy of placing recorders at set intervals along transects inherently reduced spatial bias. For the set of iterations using pooled data, we combined the two datasets by treating each acoustic sampling event as a pseudo-checklist submitted under the Stationary protocol with number of observers = 1 and duration = 1 hr (accounting for the aforementioned lumping detections into 1 hour bins to reduce pseudoreplication) [42]. Next, we determined the target species' detection frequency within the sample and, if it was < 25%, used synthetic minority class oversampling (SMOTE) [42,97] to increase the number of presences to ~25% of the total number of observations. For pooled models, the detection frequency calculation and minority class oversampling procedure was performed to eBird and acoustic data separately. Finally, in order to reduce the effect of data pooling and/or oversampling procedures on sample size differences between the three model classes, we performed the bootstrap procedure on the eBird dataset first and subsampled the data used in the acoustic and pooled bootstrap models so that all three models within the same bootstrap iteration had equal sample sizes.

Habitat predictor data were extracted for all spatial points based on raster values within circular buffers around each point. For continuous predictors, mean and standard deviation summary values were calculated within each buffer; for categorical predictors, we calculated percent coverage of each land cover class and edge density (total edge length of patches of the target class per unit area [82]). We first calculated summary statistics for a 300 m buffer radius, or approximately double the radius of the zone of acoustic detectability (150 m) for ARUs in intact forest. We believe 300 m allows for comprehensive measurement of the habitat that is relevant to our target species as it is the distance limit for normal movements of interior forest birds in the Neotropics [68]. As our eBird data contain traveling checklists of up to 1 km, we calculated summary statistics for a second set of 1 km radius buffers and performed exploratory data analysis to determine the strength of correlation between spatial scales. As many predictor summary values were not strongly correlated between spatial scales (S3 Table), we chose to retain both buffers for the encounter rate analysis. Including both spatial scales is acceptable in this step as the *randomForest* modeling framework is extremely robust against multicollinearity [98,99]. Effort covariates were also included for each observation, including year, day, and time of day as well as duration in hours, distance covered in km, observer speed in km/h (relevant for traveling checklists), and number of observers.

Encounter rate models were fit to each bootstrap's training dataset using the '*ranger*' R package and were subsequently calibrated using a constrained monotonically-increasing GAM implemented in the '*scam*' R package with uncalibrated encounter rate predictions as predictor and known site occurrence frequencies as response. Using a constrained GAM is recommended as a calibration method in the literature on eBird SDM as it reduces the chance of overfitting the model given that the calibration is being performed using the training data [18,100,101]. We produced separate training and evaluation datasets for each bootstrap iteration by splitting the data into 80%-20% partitions with positive and negative classes split separately to ensure that at least one presence and absence were included in each set. As the goal of this analysis was to evaluate models based on how they performed across the entire survey area, we ensured that all bootstrap models were evaluated on test sets containing both eBird and acoustic data. Models were evaluated using i)

Cohen's Kappa, which measures overall performance in predicting presence and absence based on the test set [102]; ii) sensitivity – the probability that a given presence prediction represents a true presence; iii) specificity – the probability that a given absence prediction represents a true absence; iv) Mean-Squared Error (MSE), which measures the mean of the squared deviations between predicted probability of presence and true presence-absence state; and v) the Matthew's Correlation Coefficient (MCC), which combines model precision, negative predictive value, sensitivity, and specificity [103]. To examine overall patterns of model predictive accuracy, we produced raster visualizations of each species' encounter rate mean and standard deviation for the entire study area based on predictions from all 100 bootstrap runs within each model class.

Finally, we used the "importance" functionality within 'ranger' to calculate relative predictor importance within each model. We chose to measure predictor importance using the Gini impurity score implementation [104], which describes the mean decrease in model accuracy when the chosen predictor is excluded during the fitting process [82]. While the impurity score calculation is highly computationally efficient, it is important to note that Gini impurity scores can suffer from importance dilution when predictors are highly correlated [105]. Our decision to use a bootstrapping procedure necessitated making this tradeoff; however, other researchers seeking to replicate these methods may be better served by using the permutation importance method, which is slower but somewhat more robust against multicollinearity [101]. Our inclusion of predictors from both 300 m and 1 km spatial scales allowed us to directly test whether one spatial scale was preferred over another in each of the model classes. This was done by fitting a linear mixed-effects model [106] with importance score as response, predictor and the full interaction between distance and dataset as fixed effects, and species as a random effect. We also used these scores to test two predictions about relative importance shifts of habitat predictors between model classes. First, as we view the encounter rate metric to be the product of a "signal" (the occurrence process) which we care about and "noise" (the detection component) that we wish to ignore, we value models in which habitat predictors are assigned high ranks relative to detection covariates. As the acoustic dataset has more consistent levels of survey effort than the eBird dataset, we predicted that habitat predictors would be ranked more highly relative to detection covariates in the pooled models than in the eBird-only models. We tested this prediction using a linear mixed-effects model with importance as response, predictor as a fixed effect, the full interaction of "is habitat predictor" (categorical, yes or no) with dataset as a second fixed effect, and species as a random effect. Second, as we felt acoustic data provide more specific information than eBird data about where species occur within intact forest, we predicted that certain forest characteristic indicators (FCIs) should be ranked higher in importance relative to predictors measuring occurrence differences between forest and non-forest areas in the pooled model class compared to the eBird-only model class. We designated HAND and percent cover of floodplain, transition forest, and terra firme as our FCIs: HAND is the single strongest proxy of forest character in this region as flooding disturbance frequency follows an elevation gradient away from major rivers [58], and the two land cover variables are both very strongly correlated with HAND ($r \geq 0.8$). As such, we feel species responses to these predictors should strongly indicate in which part of the intact forest landscape they occur. By contrast, we considered canopy height (not strongly correlated with HAND, $r \sim 0.4$), percent cover crops, percent cover built area, and percent cover bare ground to be more relevant to differences in species occurrence *between* forest and non-forest areas. We assessed this relationship using a linear mixed-effects model with importance as response, predictor as a fixed effect, the full interaction of "is FCI" (categorical, yes or no) with dataset as a second fixed effect, and species as a random effect.

## Occupancy modeling

Occupancy modeling involves jointly modeling occurrence (ψ) and detection probability (p) as separate processes based on relevant subsets of predictor variables [20]. Similar to the encounter rate models, we filtered this dataset to contain only complete checklists with survey duration <5 hours, effort distance <1 km, and number of observers <5. eBird data can be used for occupancy modeling, but in order to meet the standard set of occupancy model assumptions, it requires special processing to select a subset of the data that represents repeat visits by the same observer to the same site within

periods of closure over which species presence is believed to be constant [20,82]. We used the function "filter_repeat_visits" in the *auk* R package [81] to generate a subset of the eBird dataset containing between 2 and 10 repeat visits from the same observer to the same site within a period of 90 days, a period approximately equal to half the length of the two climatological seasons in the southwestern Amazon. Although we are not aware of any literature evidence documenting significant seasonal shifts in habitat utilization for our target species, this has been documented for other species of birds at Los Amigos between peak dry season and peak wet season [62], and we wished to have one closure period for each of these peak periods in addition to one each for the two transitional periods centered on November and April. To account for close spatial clusters of eBird points (typically seen when personal locations are submitted near but not at an eBird hotspot), we rounded all coordinates to 2 decimal points (~1.1 km at this latitude) in the filter function. Spatiotemporal sub-sampling was again applied to eBird data using the same process used for the encounter rate models. Acoustic data were split into 90 day periods of closure with repeat visits defined as consecutive 3 day-long periods of recording time [80], but were not spatiotemporally subsampled. Data were formatted into site-state matrices using functions from the R package '*unmarked*' [107]. We calculated the amount of anthropogenic habitat within each buffer size class as the sum of "crop" and "built area" land cover types as these are the two classes that are unambiguously the result of human influence. This accounts for residual differences in occurrence due to human land use that are not explained by canopy height alone. At this stage, we randomly selected 20% of presence sites and 20% of absence sites to hold out as part of an evaluation set, ensuring that both sets contained proportional samples of sites from natural and anthropogenic sites and, for the pooled models, of acoustic and eBird sampling sites. All predictors were mean-centered and standard deviation-scaled as part of the model formula specification.

Occupancy models were constructed using two single-species occupancy frameworks in the *spOccupancy* R package, both of which are based on modeling Pólya-Gamma latent variables using an MCMC sampler [108]. Audio-only and eBird-only models were fit using the simple occupancy model function "PGOcc," while the pooled model was fit using the integrated modeling function "intPGOcc," which calculates joint occurrence probabilities using data from two or more datasets with potentially different detection processes. Occupancy models are extremely sensitive to collinearity, so it was necessary to implement a comprehensive model selection process to select optimal predictor sets. Based on our exploratory data analysis and an examination of preliminary results from the encounter rate modeling component, we allowed audio-only and pooled models to contain habitat predictors from both 300 m and 1 km spatial scales, while only including 1 km predictors in the eBird-only models. For detection covariates, we allowed the eBird-only models to contain a complement of typical eBird checklist effort covariates (day of year, hours of day, effort hours, protocol type (stationary or traveling), effort distance (km), and number of observers). For the audio-only models, we included the subset of these covariates that are relevant to ARU data (day of year, hours of day, and effort hours). To account for the fact that site-surveys represent multiple days in the ARU data, we defined

hours of day as representing the hour in which the first detection of the given species occurred during a sampling period, and effort hours as the total number of hours during each three day survey period that the recorder was turned on. For each model class, we removed all predictors with non-zero variances as they have no intrinsic meaning for the given localities (e.g., anthropogenic habitats among the interior forest acoustic sites) and selected non-correlated predictors using a hierarchical clustering approach [109]. Within each cluster of highly correlated predictors ($|r| < 0.7$), we selected the predictor with the highest mean importance score in the encounter rate model for the corresponding species and model class. We then performed a multi-stage model selection procedure that has been used by other researchers fitting occupancy models with large sets of predictors using *spOccupancy* [110]. In summary, we fit each predictor (as both linear and quadratic terms to account for possible unimodal responses) to the training set as a univariate model, with an intercept-only detection formula if the predictor was an occurrence predictor and vice versa if the predictor was a detection predictor. We considered any predictors with a 95% Bayesian Credible Interval (BCI) that did not overlap zero in its respective univariate model to be unambiguously informative. All predictors that were selected in this step were entered into a global model that was subject

to backwards stepwise elimination until no additional improvement to Widely-Applicable Information Criterion (WAIC; an extension of AIC that is better suited to systems with latent variables [111]) was observed upon removing more predictors. Predictors in the best model were considered significant if their 95% BCI did not overlap zero, and any that were present in the best model while not being significant on their own were considered minorly informative. Each species-model class combination was fit 10 times using different samples of training data to ensure that results were broadly generalizable. All modeling was performed using default initial values and vague Gaussian priors for beta coefficients with $\bar{x} = 0$ and $\sigma^2 = 2.72$. Models were run for three chains, each with 28000 iterations, a burn-in period of 3000 iterations, and a thinning rate of 0.25, and model convergence was assessed by ensuring $R$-hat values for all parameters were $< 1.1$.

To assess model fit and compare different model classes, we calculated WAIC for the best model for each bootstrap. We compared model WAIC to the WAIC value of a model with intercept-only occurrence and detection formulas (hereafter "null" model) fitted on the same training data to gauge whether models encoded meaningful information. We also chose to evaluate model performance on an independent evaluation dataset, as this provided a better assessment of model generalization and allowed us to conduct direct comparisons between the three model classes. The core of this evaluation set was the 20% hold-out set from model training; however, we chose to add the set of eBird sites that were removed during the spatial subsampling component of training to ensure that we would still have sufficient numbers of presence and absence points after subsequent filtering steps. Spatiotemporal subsampling was applied to the eBird evaluation sites using the same process as was used during training; we did not spatially subsample the acoustic sites. We extracted our suite of environmental and effort data for site visits and, to match the format of the training data, we filtered visits so that eBird site detection histories represented a random sample of 2–10 repeat visits by the same observer within a period of 90 days and acoustic data detection histories represented 3 day-long periods of recording time.

When evaluating occupancy model performance on new sites, it is important to be reasonably confident that site absences are due to non-occurrence rather than non-detection, especially in situations when survey effort is highly variable across sites [43]. Therefore, we used each species' single data source models to predict cumulative detectability across all chosen visits using the formula:

$$p_i = 1 - \prod_{t=1}^{n} 1 - p_{it}$$

Here, $p_i$ is the $i$th site, $t = 1 \ldots n$ is the $n$th visit to that site, and $p_{it}$ is the estimated detection probability for each site-visit. Sites with low predicted levels of detectability ($\leq 0.9$) were removed from the validation set [43]. Although acoustic sites had substantially higher sampling intensity on average than the eBird sites (mean total effort-hours of 176 per site versus 3.6 for the eBird sites), exploratory data analysis nonetheless showed that cumulative detectability at a small number of acoustic monitoring sites with the shortest total recording durations was $\leq 0.9$.

The resulting datasets were heavily imbalanced in favor of the eBird sites in all species, so we bootstrapped model evaluation by calculating a set of model metrics (precision-recall AUC, sensitivity, and specificity) on 100 subsamples of the evaluation set that contained all acoustic sites and an equal number of randomly chosen eBird sites with representative ratios of presence and absence sites and reported mean values of the model metrics across all bootstraps. Model coefficients, confidence intervals, and effect size estimates were extracted for comparison between model classes. We also visualized model predictions by producing raster maps of occupancy probabilities for each species-model combination across the study area. Finally, we produced occupancy raster maps by taking the mean and standard deviation of the rasters from each of the 10 bootstrap runs.

### Research ethics and organismal impacts

Work was conducted with permission from the *Servicio Nacional Forestal y de Fauna Silvestre* (SERFOR, permit # D000117-2023-MIDAGRI-SERFOR-DGGSPFFS-DGSPFS). As acoustic monitoring was conducted at remote sites in the

rainforest away from populated areas, we are confident that no audio of human speech or specific human activities was collected. Minor exceptions include distant motor noise of boats traveling on the river and dredgers operating at mining sites >2 km away from the station. All data collection was passive, and we do not have any evidence that autonomous recording devices influenced the behavior or well-being of organisms in any way.

## Results

In total, the training dataset included 19,160.5 hours of survey audio, 1216.5 from the 2019 field season, 4575 from the 2020 field season, and 13,368 from the 2024 field season. We recorded 768 independent detections for *F. analis* (4.01% of recording hours had at least one detection), 313 for *M. campanisona* (1.63% of recording hours), 272 for *O. salvini* (1.42% of recording hours), 158 for *A. goeldii* (0.83% of recording hours), and 132 for *F. colma* (0.69% of recording hours). The eBird dataset included 5034 total presence-absence points, of which 444 represented presence records for *F. analis*, 146 for *A. goeldii*, 42 for *F. colma*, 40 for *O. salvini*, and 39 for *M. campanisona* (all numbers represent counts after filtering for effort covariates but prior to applying SMOTE and/or spatiotemporal subsampling).

In the encounter rate analysis, although all species-model class combinations had somewhat mediocre precision-recall AUC and sensitivity scores, the pooled models measured higher than eBird-only models for almost every metric, and in certain cases also measured slightly higher than the audio models. Mean Kappa and MCC across all model bootstraps were uniformly higher in the pooled models compared to the eBird-only models (Fig 2). While the summary performance metrics of the audio-only models are less comparable to those of the other two model classes than the two are to one another as the former model class does not contain data from the entire survey area, pooled models for the majority of species scored higher in terms of Kappa (4/5 species) and MCC (3/5 species) than the audio-only models, while the eBird-only models always scored worse than the audio-only models for these two metrics (Fig 2). The pooled models had the highest precision-recall AUC for all species except *M. campanisona*, where the acoustic-only model scored higher (Fig 2). All species had models with relatively low MSE values, with the worst-performing model in this regard (*F. analis*, pooled model) having mean MSE < 0.1. However, the eBird-only, audio-only, and pooled models broadly overlapped in MSE for the majority (3/5) of species, while in the remainder the MSE of the pooled models was higher than the other two model types (Fig 2). Examining the two components of precision-recall AUC indicated that, relative to the corresponding eBird-only models, sensitivity was higher in the pooled models for all species (Fig 2), often by a substantial margin (150%−1300%, Fig 3); while specificity decreased in the pooled models for all species relative to the eBird-only models, these decreases were much more minor (Fig 2, 3). Additionally, the acoustic-only model had higher sensitivity than the eBird-only model in all five species, with the opposite true for specificity (Fig 2).

We found that 1 km summary predictors were ranked as more important on average in the eBird-only (p < 0.0001) and pooled (p = 0.0001) models, while 300 m and 1 km summary predictors were ranked about equally in the audio-only models (Fig 4a). While habitat predictors were ranked more important on average than detection covariates in all model classes (Fig 4b), we found no evidence that any of the model classes was superior at assigning a higher rank to these predictors relative to the detection covariates. Our FCIs were assigned approximately the same mean importance in all three model classes (Fig 4c). The non-forest habitat predictors were assigned a significantly higher importance in the eBird-only models compared to the audio-only models (p = 0.0062) and marginally significantly higher than the pooled models (p = 0.0719), while the pooled models assigned them approximately the same rank as the audio-only models (Fig 4c).

Having access to the predictor importance scores also allowed us to assess the negative contributions of different anthropogenic habitats to predicted occurrence of our target species. Among the various land cover types we use as predictors in the encounter rate analysis, we consider two of these, percent cover built area and percent cover crops, to be unique signals of anthropogenic disturbance. Although neither were ranked high enough to be included in the top 10 most

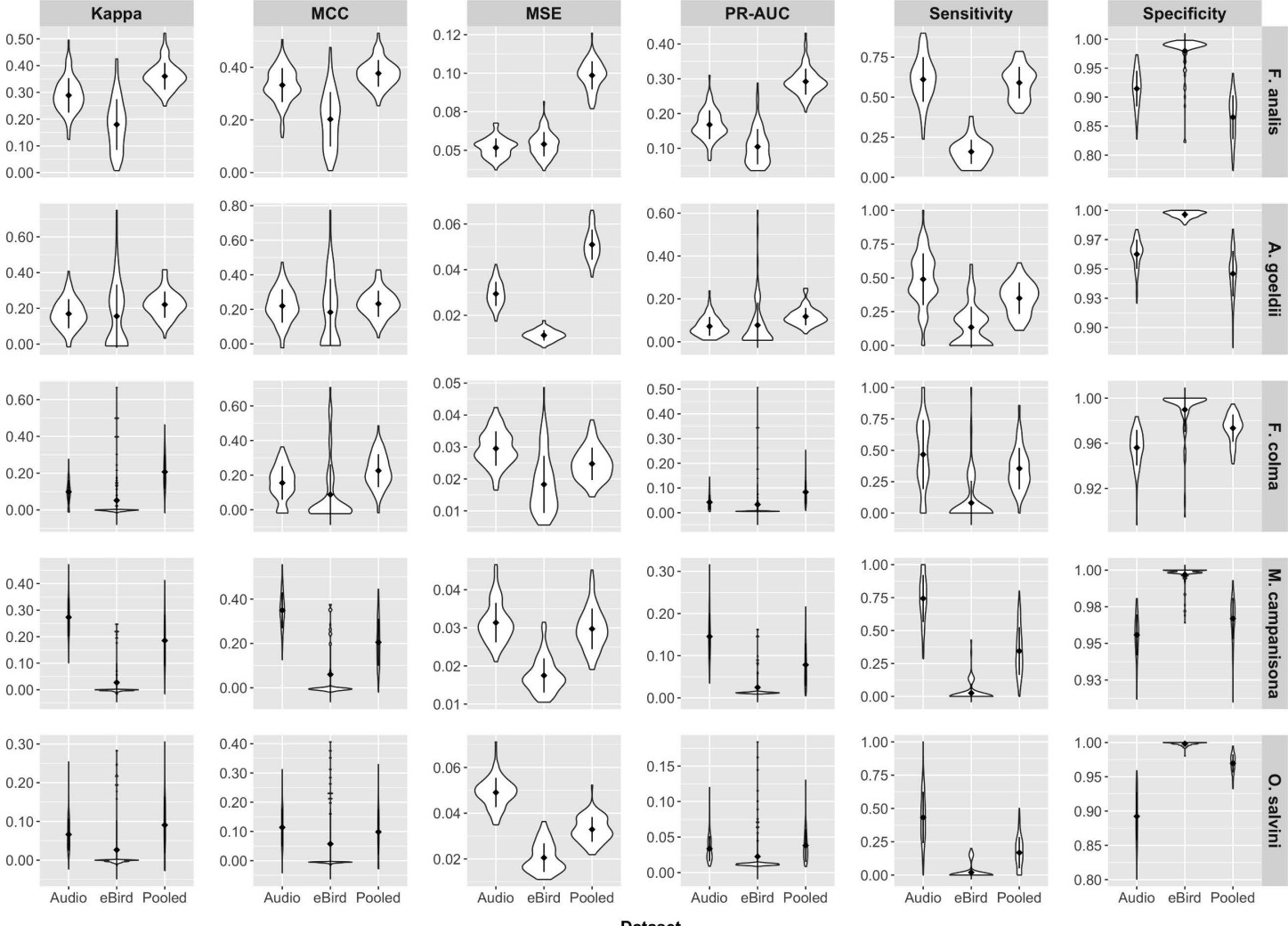

**Fig 2. Encounter rate analysis: Kappa, MCC, MSE, Precision-Recall AUC, Sensitivity, and Specificity for each model class.** Bootstrap n = 100. Higher values are better for Kappa, MCC, Precision-Recall AUC, Sensitivity, and Specificity, and lower values are better for MSE. Pooled models outperformed eBird-only models for the majority of species in all metrics except specificity.

important predictor variables for any species (S6 Fig), the former was the highest ranked "anthropogenic" land cover class in all species in both eBird and pooled model types (Table 2).

When visualizing model predictions across the study area, certain desirable properties of the pooled encounter rate models became apparent. Audio-only models tended to give plausible predictions in regions with intact forest, but failed to account for absence from anthropogenically modified habitats as this habitat type is absent in the area where acoustic data was collected (Fig 5). By contrast, eBird-only models correctly predicted absence from human-disturbed habitats, but their performance in intact forest areas often did not conform to known habitat preferences of the target species (Table 1), predicting floodplain species occurrence in terra firme habitats and vice versa (Fig 5). The pooled models balanced these performance characteristics, giving plausible predictions in both intact forest and human-disturbed habitats (Fig 5).

In the occupancy modeling analysis, all model bootstraps had lower WAIC than the null model, leading us to believe that they encode meaningful information (S4 Table). Precision-recall AUC was high (>0.7) for all species except *M.*

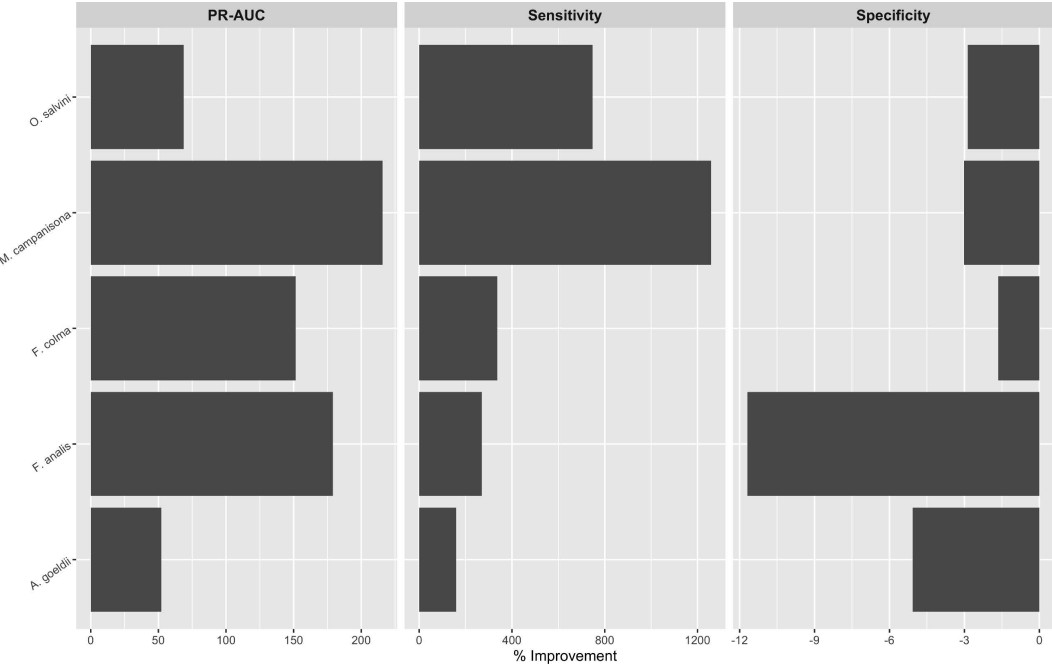

**Fig 3. Percent difference in Precision-Recall AUC, sensitivity, and specificity for pooled models relative to eBird-only models.** Precision-Recall AUC and sensitivity were higher in all species (positive values); decreases in specificity, present in all species, were more minor (negative values, note differences in x-axis scales between panels).

*campanisona*, and pooled occupancy models had higher mean precision-recall AUC scores compared to eBird-only models for 3/5 species (Fig 6). In the remaining two species, the distribution of the bootstrap scores broadly overlapped one another. In the latter case, breaking down precision-recall AUC into its components indicated that the observed differences represented a tradeoff between sensitivity and specificity (Figs 6, 7). In our combined dataset, the correlation between Canopy Height and our derived percent anthropogenic cover class was $r = -0.499$, supporting our decision to include the latter as a predictor variable.

Pooled ($\bar{x} = 3.2$ species with at least one predictor within each category, $\bar{x} = 1.9$ species with at least one significant predictor within each category) and audio-only models ($\bar{x} = 2.9$ total, $\bar{x} = 1.6$ significant) achieved slightly better representation of their respective possible predictor sets than the eBird-only models ($\bar{x} = 2.5$ total, $\bar{x} = 1$ significant) across all model bootstraps, particularly in the case of the continuous and vegetated predictor categories (Table 3, S2 File). The most frequently represented non-vegetated predictor category was open water, which is present ≤ 1 km from some acoustic sites and therefore appeared with regularity in those models. For the remaining two non-vegetated predictor categories that were allowed to appear in all three model classes (bare ground and sparsely vegetated), both eBird and pooled model classes had better representation of these predictors than the audio-only models (Table 3, S2 File). Predictors pertaining to the derived anthropogenic habitat class only appeared in two species (*F. analis* and *A. goeldi*), and were included in approximately as many bootstraps in the pooled models as in the eBird-only models (Table 3, S2 File). The top model bootstrap for each species (S3 File) generally produced plausible predictor estimates, with *F analis* site occupancy being positively associated with intermediate mean canopy height, *A. goeldi* being positively associated with increasing canopy height and edge density of transition forest, *F. colma* being positively associated with increasing HAND and decreasing floodplain edge density*, and *M. campanisona* being positively associated with increasing HAND. Less in line with expectations, the top model for *O. salvini* (pooled) predicted a positive association with extreme values of floodplain edge density (S3 File),

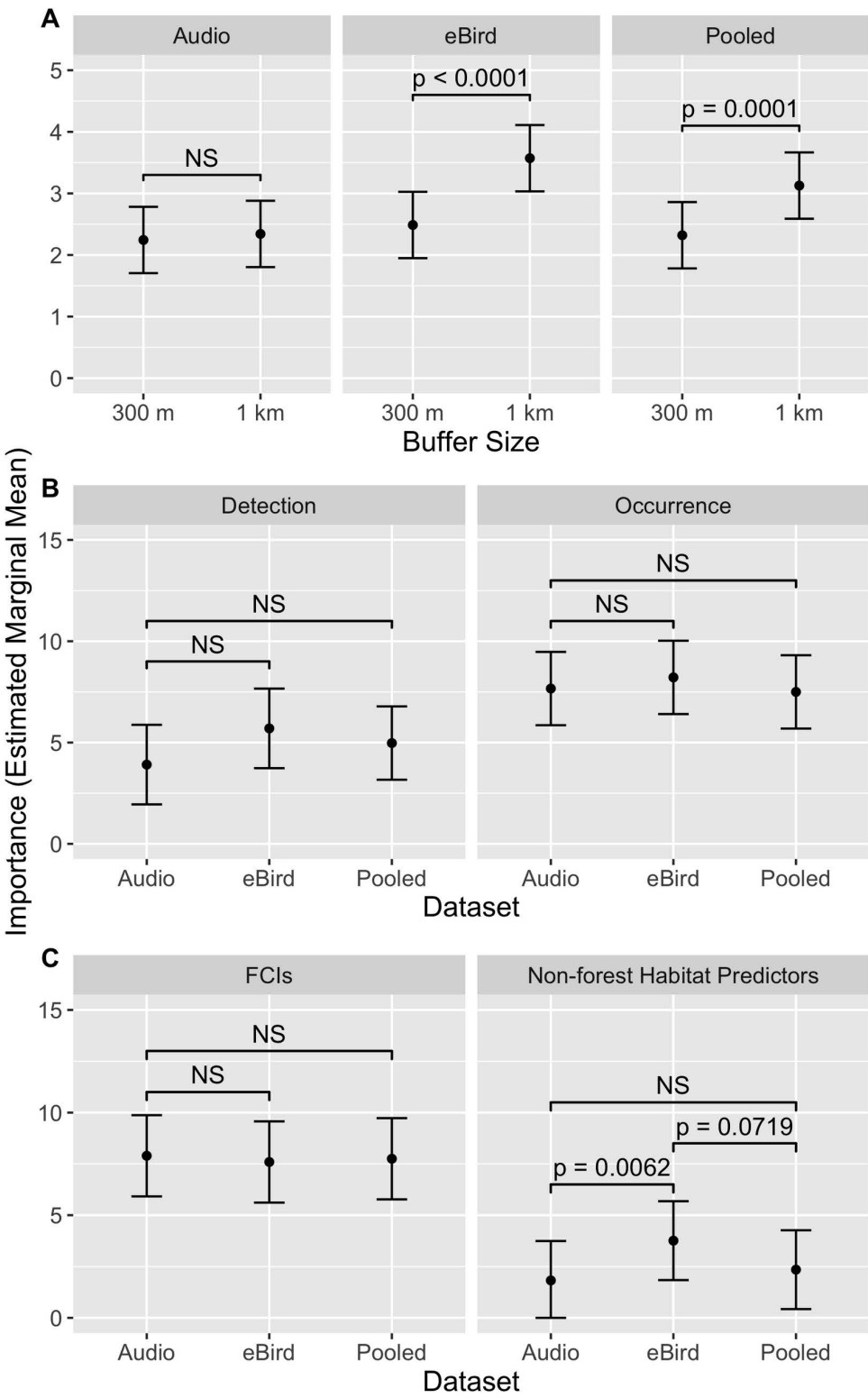

**Fig 4. Comparison of key predictor importance ranks between encounter rate model classes.** (A) 1 km predictors were ranked more highly on average in the eBird-only and pooled models, while 300 m and 1 km predictors were ranked equally in the audio-only models. (B) Habitat predictors were ranked more highly on average than detection covariates in all model classes, but no model class was superior at assigning a higher rank to these

predictors relative to the detection covariates. (C) FCIs ranked about equally in all three model classes. Non-forest habitat predictors were assigned a significantly higher rank in the eBird-only models compared to the audio-only models and marginally significantly higher than the pooled models, but about equal in audio-only and pooled models. For computational efficiency, values represent estimated marginal means of Gini impurity scores across all model bootstraps (n = 100); note that importance values may be affected by importance dilution due to multicollinearity.

but for this species neither audio-only nor eBird-only models predicted any significant predictor associations as alternatives. With the exception of predictors related to open water, none of the top model bootstraps for any species included anthropogenic or non-vegetated habitat predictors, though positive associations with canopy height were present in many of the model bootstraps.

Visual analysis of model predictions told a broadly similar story to the parameter estimates. In general, each species' audio-only model matched known habitat preferences in intact forest areas, with terra firme species predicted to occur most frequently in terra firme habitats and vice versa for floodplain species. However, as this model class had no information about the occurrence patterns of target species away from acoustic survey sites, they failed to predict absence from non-vegetated or anthropogenically modified areas (S7–S11 Figs). In contrast, and despite seldom containing any predictors associated with the "anthropogenic" land cover class, eBird-only models generally predicted absence from non-vegetated areas and anthropogenically-modified areas. However, similar to the encounter rate models, they performed poorly in intact forest areas (S7–S11 Figs). Pooled occupancy models, which were able to draw on both sets of distribution data during model fitting, produced plausible occupancy predictions over the greatest range of habitat types (Fig 8, 9).

## Discussion

Our results broadly support previous work showing that small amounts of structured survey data collected over important ecological gradients can be combined with citizen science data to better model species distribution patterns [17,19,42,43]. The data pooling approach we implemented allowed pooled models to achieve higher validation scores and predictive accuracy for most species in both the encounter rate and occupancy modeling components. Inclusion of eBird data in the pooled encounter rate models, while incurring a minor predictive accuracy penalty in natural habitats relative to the acoustic-only models, allowed the models to gain insight about species responses to non-forest land cover types. They

**Table 2. Predictor importance of anthropogenic habitat types in the eBird and pooled models for all species.** Scores are unitless and are derived from the Gini index. For each species-model combination, the bolded number indicates the predictor with the higher mean importance score. For all five species, both model types indicated that percent cover built area was the most important anthropogenic habitat type influencing species presence, though in this analysis other non-anthropogenic land cover types were assigned much higher importance scores and neither percent cover built area or percent cover crops made the list of top 10 most important predictors for any species. Only results for 1 km predictors are shown as 1 km ranks were higher than 300 m for both predictors in all species.

| | Predictor | Importance | |
|---|---|---|---|
| | | eBird | Pooled |
| *F. analis* | Percent Cover Built Area | **3.50** | **1.70** |
| | Percent Cover Crops | 1.10 | 0.7 |
| *F. colma* | Percent Cover Built Area | **2.95** | **0.67** |
| | Percent Cover Crops | 0.70 | 0.24 |
| *A. goeldi* | Percent Cover Built Area | **4.03** | **1.70** |
| | Percent Cover Crops | 0.92 | 0.65 |
| *M. campanisona* | Percent Cover Built Area | **1.77** | **0.83** |
| | Percent Cover Crops | 1.07 | 0.65 |
| *O. salvini* | Percent Cover Built Area | **2.98** | **1.27** |
| | Percent Cover Crops | 2.51 | 0.76 |

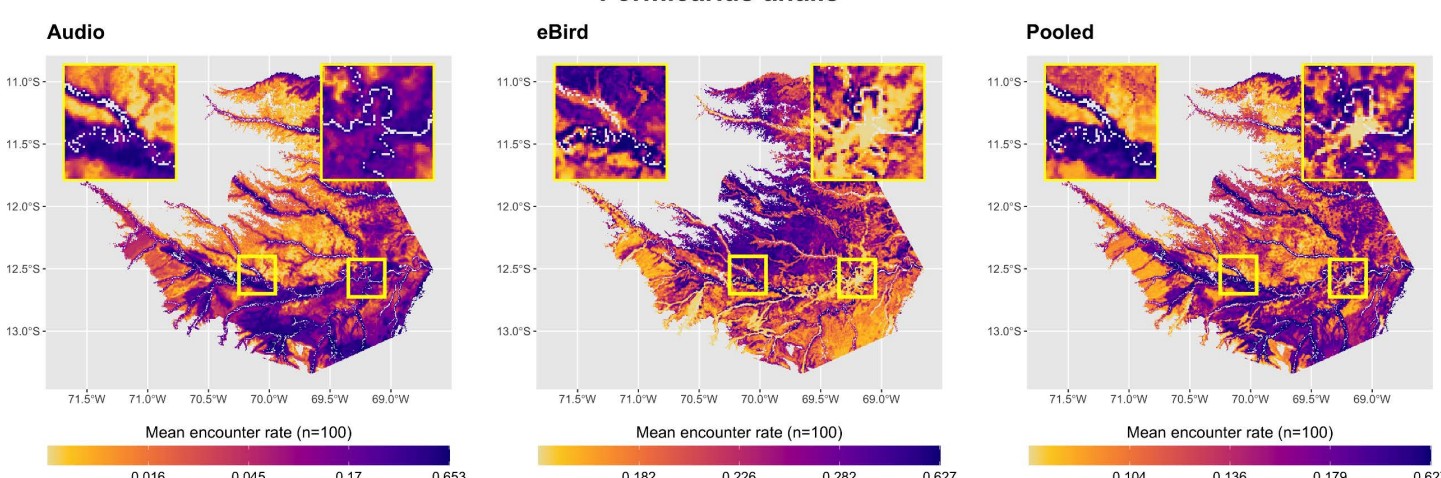

**Fig 5. Model predictions of mean encounter rate across the study area for *F. analis*, all model classes.** Inset maps show intact forest areas around acoustic monitoring sites at Los Amigos Biological Station (left) and degraded habitats in and around Puerto Maldonado, the largest urban area in Madre de Dios (right). River corridors are illustrated as white raster pixels on all maps. Of the three model classes, only pooled models achieved reasonable species-habitat associations in both natural areas (in this case, predicting high levels of *F. analis* occurrence near river channels and low values in upland areas) as well as absence from degraded and deforested habitats. Visualizations for all species in S1–S5 Figs.

had the highest average PR-AUCs for most species, and also produced the most accurate maps (Figs 2, 5). Improvement to the occupancy models was more modest from a quantitative perspective, and the pooled models were not always better than the eBird-only models for all species. Even so, the pooled models generally had better performance than the eBird-only models and were able to integrate inferences from a larger variety of spatial scales than either model class could on its own (Figs 5, 8). We feel these results are especially relevant to other researchers in this field as the small area of coverage of our audio dataset is likely representative of that which other researchers would have access to, in Amazonia or elsewhere in the Tropics. It is far more likely that these hypothetical researchers would have access to acoustic data from a single station's trail network, for instance, than one that covered a regional spatial extent. We believe that our study design's targeting of the region's most important environmental gradients is responsible for its substantial improvements in model performance despite its small spatial extent relative to the eBird dataset.

A general conclusion in this analysis is that using eBird data alone for modeling is particularly challenging in regions such as the Amazon that have high levels of beta diversity and relatively few, narrow transportation corridors relative to their land area. Low availability of eBird data away from these corridors can give rise to unrealistic model predictions, particularly when position along the habitat gradient is directly correlated with distance from transportation corridors, as is the case in this (Fig 1) and many other parts of the Amazon. Another important eBird data quality issue in this region is that records that are submitted to hotspots often actually represent birding effort distributed across a network of trails covering a larger area. In such cases, these checklists may contain a mixture of terra firme and floodplain data. The coordinate point used for a hotspot is therefore often not representative of the habitat in which data collection occurred, which makes using eBird data to predict species responses to the short, steep ecological gradients common in this region more challenging. Our restriction of eBird data to checklists collected within 1 km of their recorded coordinates and (for the occupancy models) lumping clusters of eBird personal observations with nearby hotspots reduces but does not eliminate this bias. While more restrictive cutoffs would have allowed us to further reduce bias, reducing them enough to match the spatial specificity of acoustic monitors (which in tropical forest can be placed as close as 150-200m apart while remaining independent surveys [21]) would have decreased the total sample size of our eBird dataset by over half (S12 Fig), and

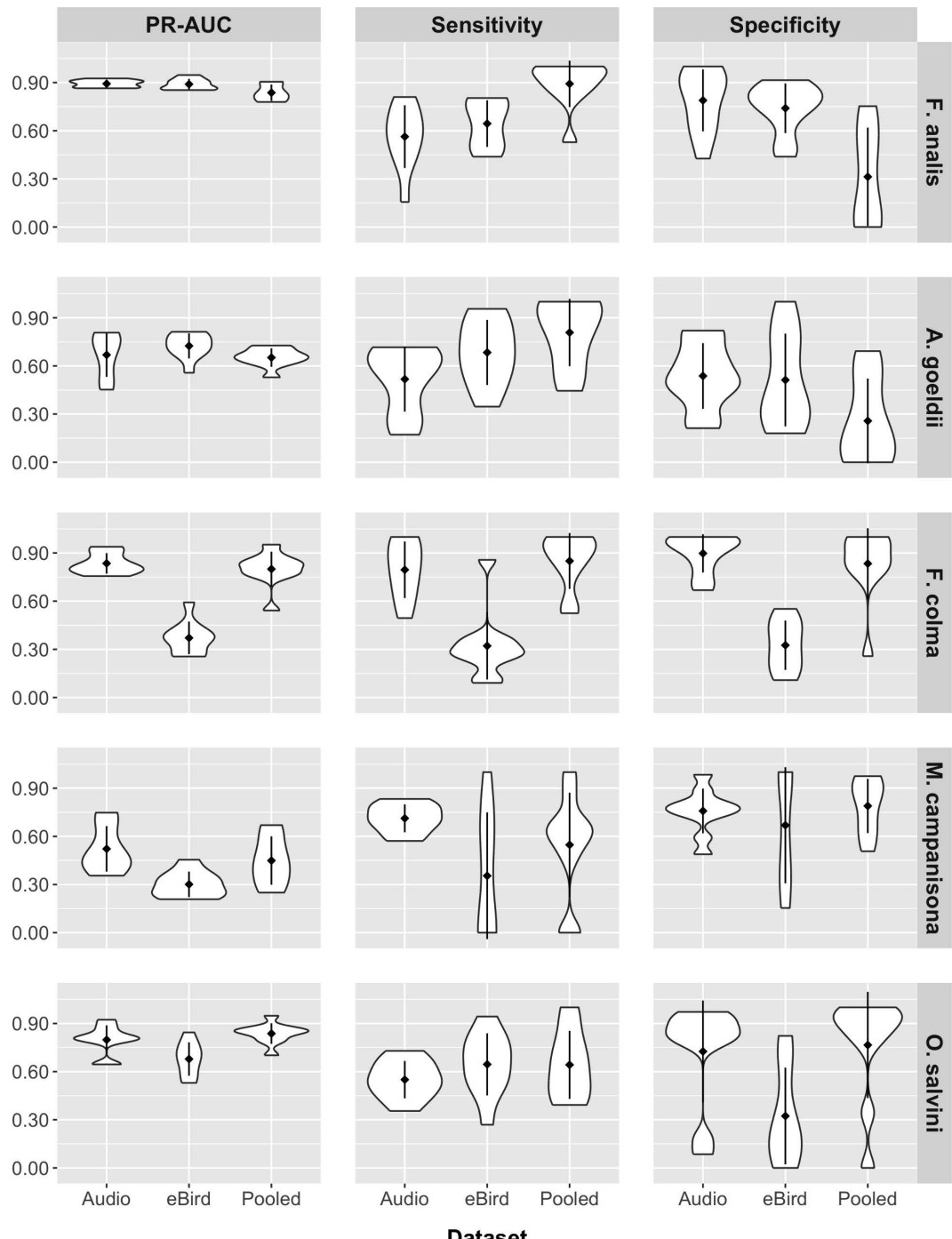

**Fig 6. Occupancy analysis: Precision-Recall AUC, Sensitivity, and Specificity for each model class.** Bootstrap n = 10. Higher values indicate better performance. Pooled models outperformed eBird-only models for the majority of species in all metrics. Precision-recall AUC was higher in the pooled models than the eBird models for 3/5 species and slightly worse in the other two.

possibly more within individual low-prevalence species. While this issue will decrease as more data accumulates over time (and we strongly suggest future researchers consider using a shorter distance cutoff in their analyses if data density in their system allows it), it is nonetheless an important consideration given current data density in the global Tropics.

Finally, it is worth keeping in mind that eBird data are collected by a variety of different observers that have varying degrees of experience with the local avifauna and follow different patterns of sampling effort allocation. One specific instance where this might have impacted our results was the case of the best-fit pooled occupancy model for *O. salvini*, which indicated a positive association with extreme values of floodplain edge density (S3 File) that is biologically implausible given what is known about this species' ecological preferences [75]. It appears that a few outlier site-visits from the eBird dataset are primarily responsible for driving the shape of this relationship, all of which are checklists from popular birding guides at large, popular ecolodges (S13 Fig). Calibration indices that evaluate eBird observers using species accumulation curves are a promising strategy to account for some of this variability in birding skill level and have been employed in other similar eBird SDM analyses [39,112]. Other researchers have also found success in using statistical models to compare eBird checklist observations with point counts from experienced observers to select a subset of high quality checklists for analysis [113].

Given that modeling results sometimes differed among species, we feel it is worth exploring qualitatively whether species with similar habitat preferences (floodplain species: *F. analis*, *A. goeldi*; terra firme species: *F. colma*, *M. campanisona*, *O. salvini*) showed commonalities in model performance changes. In the encounter rate component, while decreases in specificity were relatively minor across the board in the encounter rate analysis, the three terra firme species showed the smallest drops, which were paired with larger increases in sensitivity (Fig 2, 3). In the occupancy model component, the pooled models also showed less ambiguous improvements in predictive accuracy for the terra firme species than for the floodplain species (Figs 7, 8). In context, both of these results suggest that the apparent occurrence patterns of terra

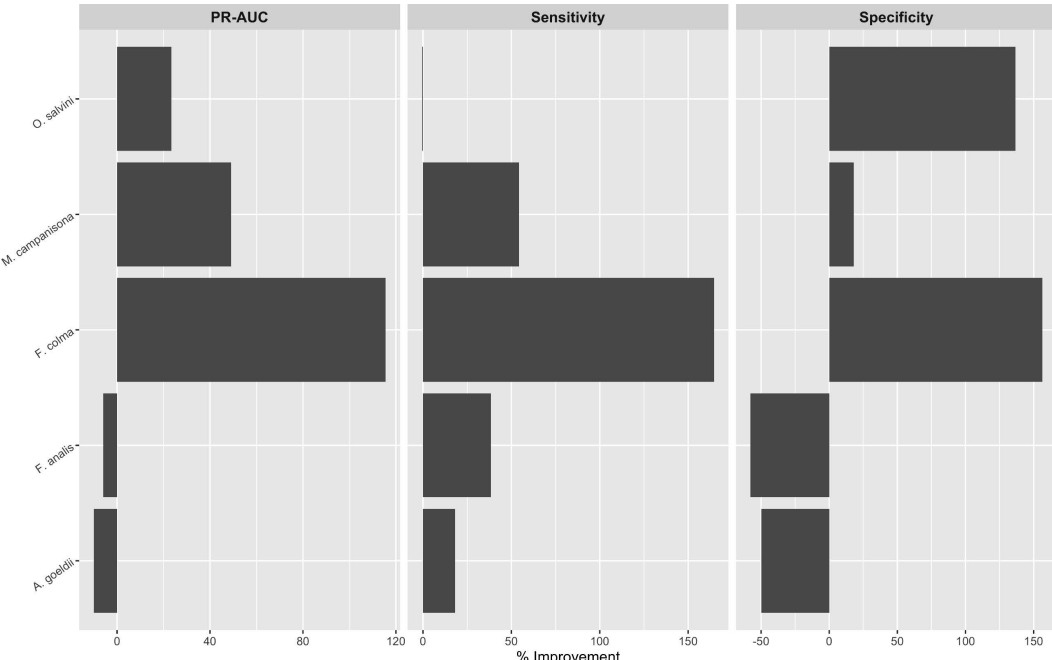

**Fig 7. Percent difference in Kappa, sensitivity, and specificity for pooled occupancy models relative to eBird-only models.** The pooled models of 3/5 species had higher overall model AUC as compared to their corresponding eBird-only models. Where AUC of the pooled models was lower, this appeared to be due to tradeoffs between sensitivity and specificity.

Table 3. Parameter frequencies across all occupancy model bootstraps. Values in the "Total" and "Sign." columns represent the relative frequency with which any variant of the predictor specified in the column header (encompassing quadratic variants, mean, SD, percent cover/ edge density, and predictors at both 300 m and 1 km scales) was included or was significant (95% BCI not containing zero) in the model bootstraps, respectively. Values are expressed as proportions relative to the maximum number of possible predictors across all model bootstraps to account for differences in the total number of possible predictors in each model class, with darker cell colors indicating higher numbers. Pooled and audio-only models generally had greater representation of predictors from the continuous and vegetated categories than the eBird-only models. Both eBird and pooled models contained more predictors from the non-vegetated predictor types that were included in all model classes than the audio-only models.

| Species | Data-set | Continuous | | | | Vegetated | | | | | | | | Non-Vegetated | | | | | | | |
| | | Canopy | | HAND | | Flooded Vegetation | | Floodplain | | Transition Forest | | Terra Firme | | Open Water | | Bare Ground | | Built Area | | Sparsely Vegetated | |
| | | Total | Sign. | Total | Sign. | Total | Sign. | Total | Sign. | Total | Sign. | Total | Sign. | Total | Sign. | Total | Sign. | Total | Sign. | Total | Sign. |
|---|---|---|---|---|---|---|---|---|---|---|---|---|---|---|---|---|---|---|---|---|---|
| F. analis | Audio | 0.13 | 0.05 | 0.03 | – | – | – | 0.20 | 0.10 | – | – | 0.06 | 0.04 | 0.28 | 0.04 | 0.01 | – | X | X | 0.01 | – |
| A. goeldii | Audio | 0.10 | 0.08 | 0.10 | 0.06 | 0.05 | 0.04 | 0.01 | 0.01 | 0.01 | 0.01 | 0.01 | 0.01 | – | – | – | – | X | X | – | – |
| F. colma | Audio | 0.14 | – | 0.08 | 0.01 | – | – | 0.16 | 0.10 | 0.04 | – | 0.11 | 0.06 | 0.05 | – | – | – | X | X | – | – |
| M. cam-panisona | Audio | 0.15 | – | 0.19 | 0.14 | – | – | 0.10 | 0.04 | – | – | 0.01 | – | 0.26 | – | – | – | X | X | – | – |
| O. salvini | Audio | 0.05 | – | – | – | – | – | 0.06 | – | 0.01 | – | 0.04 | – | 0.01 | 0.01 | – | – | X | X | – | – |
| F. analis | eBird | 0.33 | 0.25 | – | – | – | – | – | – | – | – | 0.05 | 0.03 | 0.08 | – | – | – | – | – | – | – |
| A. goeldii | eBird | 0.13 | 0.08 | 0.03 | 0.03 | – | – | – | – | 0.13 | 0.10 | 0.10 | 0.05 | 0.03 | 0.03 | – | – | 0.05 | – | – | – |
| F. colma | eBird | – | – | 0.03 | – | – | – | 0.25 | – | 0.25 | – | 0.03 | – | 0.40 | – | 0.08 | – | – | – | – | – |
| M. cam-panisona | eBird | 0.03 | – | – | – | – | – | 0.05 | – | – | – | 0.15 | – | 0.03 | – | – | – | – | – | – | – |
| O. salvini | eBird | 0.13 | 0.03 | 0.25 | 0.03 | – | – | 0.05 | – | 0.10 | 0.03 | – | – | 0.08 | – | – | – | – | – | 0.03 | – |
| F. analis | Pooled | 0.10 | 0.05 | 0.01 | – | 0.01 | – | 0.16 | 0.15 | 0.01 | 0.01 | 0.05 | 0.03 | 0.16 | 0.01 | – | – | 0.06 | – | – | – |
| A. goeldii | Pooled | 0.06 | 0.01 | 0.05 | 0.01 | – | – | 0.08 | 0.04 | 0.14 | 0.03 | 0.03 | – | 0.05 | 0.05 | – | – | – | – | – | – |
| F. colma | Pooled | 0.33 | – | 0.16 | 0.03 | – | – | 0.21 | 0.05 | 0.19 | 0.03 | 0.11 | 0.03 | 0.06 | – | – | – | – | – | 0.03 | – |
| M. cam-panisona | Pooled | 0.16 | 0.09 | 0.13 | 0.04 | – | – | 0.01 | – | 0.01 | – | 0.05 | 0.01 | 0.09 | – | – | – | – | – | – | – |
| O. salvini | Pooled | 0.11 | 0.03 | 0.03 | – | – | – | 0.14 | 0.03 | 0.01 | – | 0.09 | – | – | – | – | – | – | – | – | – |

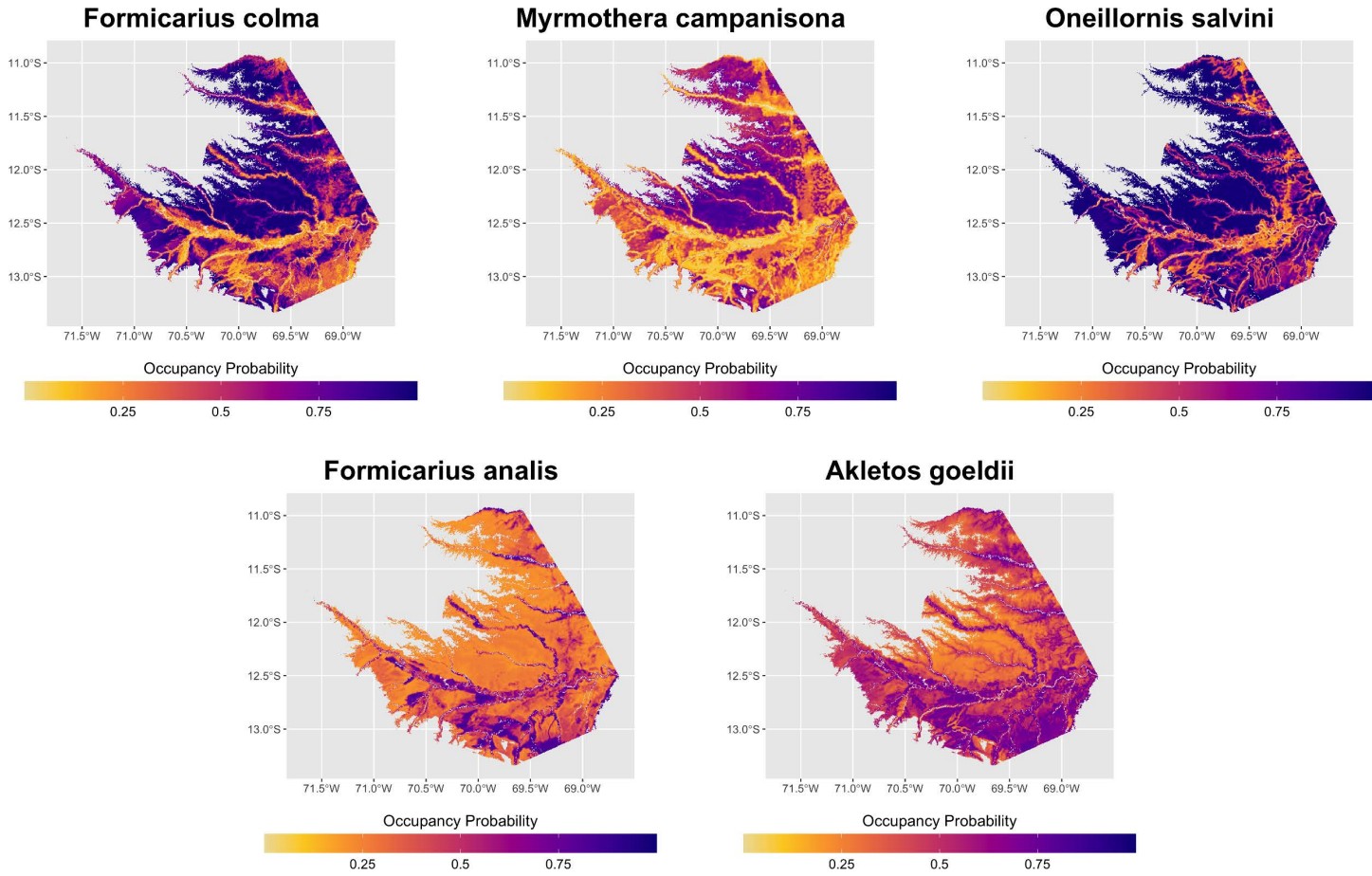

**Fig 8. Pooled occupancy plots for all target species.** Data pooling provides more realistic distribution models, with floodplain species (bottom row) showing higher predicted occurrence in floodplain and terra firme species (top row) in terra firme, respectively, along with absence from anthropogenically modified habitats.

firme species diverge more from one another between the two component datasets than they do for the floodplain species. As is generally true in the Amazon [114], eBird observations in Madre de Dios have known patterns of non-random spatial distribution biased towards the major rivers that serve as the main means of transportation in this region (Fig 1). Not only do many people eBird by boat while traveling in the region, many eBird station or lodge "hotspots" are physically located on top of ecolodge buildings, which tend to be close to rivers for logistics regions, and therefore have a HAND distribution that is biased toward the lower end of the critical floodplain-terra firme ecological gradient (S14 Fig). This could be evidence of systemic differences in data quality in the eBird dataset between habitat assemblages: near intact forest, eBird data may more accurately reflect the habitat affinities of floodplain species than terra firme species due to the fact that it is more likely to be collected in the "correct" habitat. In this case, the addition of acoustic data would be especially beneficial in the pooled models of terra firme species, as we observed to be true in our analysis (Figs 3, 7). We recommend that other researchers seeking to implement a data pooling strategy for birds in tropical forest think critically about how local patterns of sampling unevenness within their eBird dataset might differentially impact modeling performance for different habitat assemblages.

Interestingly, the model selection process we used to select informative predictors for the occupancy models only occasionally retained the dedicated "anthropogenic" land cover class we included in the dataset in the final models (Table 3).

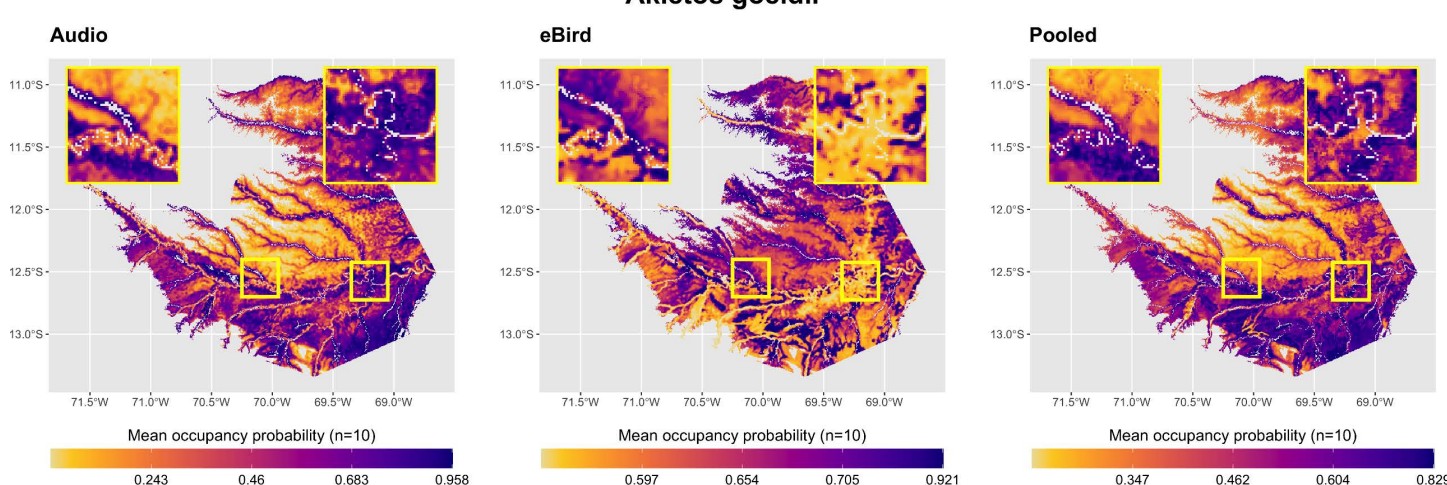

**Fig 9. Model predictions of mean occupancy probability across the study area for *A. goeldi*, all model classes.** Inset maps show intact forest areas around acoustic monitoring sites at Los Amigos Biological Station (left) and degraded habitats in and around Puerto Maldonado, the largest urban area in Madre de Dios (right). River corridors are illustrated as white raster pixels on all maps. Of the three model classes, only pooled models achieved reasonable species-habitat associations in both natural areas (in this case, predicting high levels of *A. goeldi* occurrence near river channels and low values in upland areas) as well as absence from degraded and deforested habitats. Visualizations for all species in S7–S11 Figs.

Despite this, qualitative examination of the occupancy models indicates that the models still predict non-occupancy in open and human-modified habitats (Fig 9). Canopy height and many of the vegetated land cover predictors frequently appear in the model bootstraps, and while they are not strongly correlated enough with the anthropogenic land cover category to be removed during the model selection process, they may nonetheless account for some degree of variance in habitat parameter space between vegetated and non-vegetated areas, perhaps in concert with one another rather than alone. It is probable that this allows them to serve as "stand in's" for the anthropogenic habitat predictor in the models.

Having an accessible technique for generating SDM models for Amazonian birds that recapitulates known habitat associations has important implications for understanding the distribution of tropical species. Despite being vastly more speciose than the north temperate zone, the Tropics have historically had significantly lower participation rates in eBird and other citizen science projects [115], which makes testing data integrations designed to make the most of this limited data particularly critical in this system. While our results broadly support previous work showing citizen science data and structured survey data on habitat gradients can be combined to produce better SDMs [17,19,42,43], this is the first time that we are aware of such a result having been shown for a tropical forest ecosystem. Although a handful of recent studies have used eBird data to measure the distribution patterns of tropical birds [116,117], these studies were both conducted in regions of South Asia that have higher eBird data densities than our region (Madre de Dios has 0.56 eBird checklists/ km² compared to, for instance, 3.83 eBird checklists/ km² in Tamil Nadu) and compared to the global Tropics in general [36,115], and neither used an integrated modeling approach.

We also believe the use of acoustic monitoring rather than in-person field surveys as the structured dataset for our analysis is a novel contribution to this field. Prior integrated modeling approaches using eBird data have incorporated data from The Nature Conservancy, the Breeding Bird Atlas, the USGS Breeding Bird Survey, and other similar highly structured surveys [42,43]. Few tropical countries have a significant history of conducting similar bird atlas surveys [118]; by contrast, an increasing number of biological stations have available acoustic survey data [14,15,22,119]. We argue that not only can acoustic surveying fill some of these data availability gaps by providing similar data quality for highly vocal species to that from human observations [120], they have the added benefits of being non-invasive and easier to conduct

for extended periods of time. They also have a proven history of providing critical information on poorly-known species [121–123]. In the past, the use of acoustic survey data for landscape-scale analysis has been hindered significantly by the need for manual processing of audio by trained listeners with intimate familiarity of the avian soundscape being examined [124]; however, this barrier is rapidly being overcome by automated detection techniques, either produced in-house or using publicly-available classifiers such as BirdNET [14,77].

While we show that acoustic monitoring data can be used as a stand-in for human survey data in this analysis, we caution that the two survey methods are not guaranteed to produce identical inferences. A primary difference is that observers are capable of detecting non-vocal individuals by sight, and it is probable that two models generated for a species with lower propensity for loud repetitive vocalizations, one using human survey data and the other using acoustic monitoring data, would yield different conclusions about that species' patterns of occupancy. Our analysis preferentially looks at a small set of secretive but highly vocal species for which acoustic monitoring is an optimal survey approach, but other researchers must carefully consider factors that could limit its utility for their subject species. Careful consideration must be given to potential variation in vocal activity by season, which has been reported for some Amazonian birds [125], as well as spatial variability in vocal rates across species' home ranges that is related to aspects of species life history [126]. For instance, while none of our species lek, vocal rates of lekking species are likely to vary dramatically within suitable habitat due to the non-random distribution of appropriate lekking sites, meaning that small variations in recorder site placement could yield dramatically different measures of site occupancy [127]. In either case, it is important to use existing domain knowledge about the subject species to assess whether the chosen survey design will yield "acoustic occupancy" rates that are comparable to true site occupancy. We also suggest that other researchers seeking to employ acoustic monitoring in their work on tropical bird species consider gathering a set of point-count surveys in tandem with acoustic monitoring to allow for an independent set of inferences or to directly calibrate site occupancy values, as there is recent precedent for employing this technique in temperate systems [128].

While we mostly framed anthropogenic land use categories as being representative of human disturbance near major developed areas in Madre de Dios, we want to emphasize that it is also possible for these disturbed habitats to occur along riverine corridors within otherwise intact forest areas. While neither percent cover crops nor percent cover built were ranked high enough to be included in the top 10 most important predictor variables for any species in the encounter rate analysis (Fig 4, S6 Fig), both of these land cover classes do nonetheless appear within the inset map around Los Amigos (Fig 1D). Our field experience at this site suggests that these land use categories (particularly percent cover built area) are associated with alluvial gold mining activity at many sites [15]. Future work, even analyses that focus exclusively on intact forest sites, should still consider incorporating these predictors as part of the Sentinel-2 dataset [90] as a means of assessing these forms of disturbance.

We have demonstrated here that ARU surveys combined with machine learning acoustic classifiers can be used to rapidly collect occurrence datasets for interior forest birds; when used as part of an integrated modeling framework, this data can be used to model species habitat associations across habitats. In addition to helping researchers build a more complete picture of Amazonian diversity, such inferences will also be critical for predicting responses to anthropogenic land use change, which is a key research priority given rapid regional increases in deforestation, fire, road-building [63,129,130] and illegal artisanal gold mining along riverine corridors [64,65]. Understanding species' current patterns of habitat utilization will be crucial for understanding potential responses to various anthropogenic pressures as well as helping to inform conservation efforts and acquisition of land for the establishment of protected areas.

## Supporting information

**S1 Fig. Encounter rate mean and standard deviation visualization for *Formicarius analis*, all three model classes.** *F. analis* is a floodplain species. Pooled model offered high prediction accuracy both in natural and degraded habitats. (TIF)

**S2 Fig. Encounter rate mean and standard deviation visualization for *Formicarius colma*, all three model classes.** *F. colma* is a terra firme species. Pooled model offered high prediction accuracy both in natural and degraded habitats.
(TIF)

**S3 Fig. Encounter rate mean and standard deviation visualization for *Myrmothera campanisona*, all three model classes.** *M. campanisona* is a terra firme species. Pooled model offered high prediction accuracy both in natural and degraded habitats.
(TIF)

**S4 Fig. Encounter rate mean and standard deviation visualization for *Akletos goeldii*, all three model classes.** *A. goeldii* is a floodplain species. Pooled model offered high prediction accuracy both in natural and degraded habitats.
(TIF)

**S5 Fig. Encounter rate mean and standard deviation visualization for *Oneillornis salvini*, all three model classes.** *O. salvini* is a terra firme species. Pooled model offered high prediction accuracy both in natural and degraded habitats.
(TIF)

**S6 Fig. Relative predictor importance for all species x model class combinations.** A = *F. analis*, B = *A. goeldii*, C = *F. colma*, D = *M. campanisona*, E = *O. salvini*. Environmental factors are indicated in blue; detection covariates are indicated in gray. Pooled models generally did a good job of assigning high rankings to habitat variables thought to be important to species occurrence, though relative ranks varied between species.
(TIF)

**S7 Fig. Occupancy model visualization for *Formicarius analis*, all three model classes.** *F. analis* is a floodplain species. Pooled model offered high prediction accuracy both in natural and degraded habitats.
(TIF)

**S8 Fig. Occupancy model visualization for *Formicarius colma*, all three model classes.** *F. colma* is a terra firme species. Pooled model offered high prediction accuracy both in natural and degraded habitats.
(TIF)

**S9 Fig. Occupancy model visualization for *Myrmothera campanisona*, all three model classes.** *M. campanisona* is a terra firme species. Pooled model offered high prediction accuracy both in natural and degraded habitats.
(TIF)

**S10 Fig. Occupancy model visualization for *Akletos goeldii*, all three model classes.** *A. goeldii* is a floodplain species. Pooled model offered high prediction accuracy both in natural and degraded habitats.
(TIF)

**S11 Fig. Occupancy model visualization for *Oneillornis salvini*, all three model classes.** *O. salvini* is a terra firme species. Pooled model offered high prediction accuracy both in natural and degraded habitats.
(TIF)

**S12 Fig. Distribution of the eBird data used in this analysis by distance covered after applying survey effort filters.** Using eBird data for inference across the short, steep ecological gradients in the Neotropics is challenging, especially when including travelling checklists, as the coordinate point assigned to checklists often does not accurately capture the areas covered by observers. Constraining eBird data to checklists that cover distances of >1 km distance reduces, but does not eliminate this bias, and while more restrictive cutoffs would further reduce bias, doing so generally also yields dramatic reductions in sample size.
(TIF)

**S13 Fig. Predicted Site Occupancy by Edge Density Floodplain (300m) in the *O. salvini* pooled model.** Solid line indicates the shape of the relationship predicted by the highest performing model bootstrap. The quadratic shape of this relationship is mainly driven by a cluster of eBird site-visits with edge density values >1.6 (vertical dotted line) that have high predicted site occurrence values; all of these are checklists from popular birding guides at large, popular ecolodges. Removing these points leads to a relationship that is more ecologically plausible (dotted line).
(TIF)

**S14 Fig. Distribution of eBird and acoustic monitoring data points used in this analysis by HAND after applying survey effort filters.** Many eBird hotspots are near the major rivers that serve as the main means of transportation in the lowland Amazon, causing the overall distribution of HAND within the eBird data fraction to be biased toward the lower end of the critical floodplain-terra firme ecological gradient.
(TIF)

**S1 Table. List of command-line parameters used in BirdNET classification.** All classification was performed using BirdNET v2.4.
(DOCX)

**S2 Table. Model performance, logistic regression threshold selection for acoustic detections.** The logistic GAM (with logit score as a linear term and elevation as a smoother) performed best for all species.
(CSV)

**S3 Table. Correlation patterns of habitat predictors.** Many, but not all, predictors are strongly correlated between 300m and 1 km spatial scales.
(XLSX)

**S4 Table. Model performance metrics for occupancy models.** Models are compared using the Widely-Applicable Information Criterion (WAIC), an extension of AIC that is better suited to systems with latent variables All occupancy models outperformed a null model trained on the same data.
(XLSX)

**S1 File. Acoustic monitoring field study details.**
Additional details on the manner in which acoustic monitoring data was collected.

(DOCX)

**S2 File. Occupancy model bootstrap data.** Data on the total number and number of significant predictors for all predictor variants across all of the model bootstraps.
(XLSX)

**S3 File. Parameter estimates from the top occupancy model bootstraps of all species.** Additional details about the top audio-only, eBird-only, and pooled occupancy model bootstraps from each species estimates, 95% BCIs, R-hat values, and model AUCs.
(DOCX)

## Acknowledgments

We would like to thank the staff at Los Amigos for helping to ensure our safety and success during this stage of the project, and Adrian Forsyth for his mentorship and support. We also thank Edwin Jurado who collected field data in the early part of the 2024 field season, as well as Bioacoustics at the Cornell Lab of Ornithology, the Climate Corridors team, and Robert Guralnick at the Florida Museum for graciously lending equipment. Special thanks also go to the Searcy Lab at

University of Miami and Andrew Dreelin for their feedback on the project's methodology and framing of the manuscript. None of this would have been possible without your help.

## Author contributions

**Conceptualization:** Reid Rumelt.

**Data curation:** Reid Rumelt, Arianna Basto.

**Formal analysis:** Reid Rumelt.

**Funding acquisition:** Reid Rumelt, Zuzana Buřivalová, Christopher Searcy.

**Investigation:** Reid Rumelt, Carla Mere Roncal, Arianna Basto.

**Methodology:** Reid Rumelt, Carla Mere Roncal, Arianna Basto, Zuzana Buřivalová, Christopher Searcy.

**Project administration:** Reid Rumelt, Carla Mere Roncal.

**Resources:** Reid Rumelt, Carla Mere Roncal, Arianna Basto, Christopher Searcy.

**Software:** Reid Rumelt.

**Supervision:** Reid Rumelt, Carla Mere Roncal, Arianna Basto.

**Validation:** Reid Rumelt, Carla Mere Roncal.

**Visualization:** Reid Rumelt.

**Writing – original draft:** Reid Rumelt, Carla Mere Roncal, Arianna Basto, Christopher Searcy.

**Writing – review & editing:** Reid Rumelt, Carla Mere Roncal, Arianna Basto, Zuzana Buřivalová, Christopher Searcy.

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
