## [Decision Letter · Decision Letter 0]

PONE-D-24-56728Combining acoustic survey and citizen science data yields enhanced species distribution models for tropical rainforest birdsPLOS ONE

Dear Dr. Rumelt,

Thank you for submitting your manuscript to PLOS ONE. After careful consideration, we feel that it has merit but does not fully meet PLOS ONE’s publication criteria as it currently stands. Therefore, we invite you to submit a revised version of the manuscript that addresses the points raised during the review process.

The reviewers have multiple questions about the methods and results that require clarification, and they also have multiple suggestions for improvements in the writing of the manuscript. Please ensure that in your response to reviewer comments, you clearly address each comment from each reviewer. 

We look forward to receiving your revised manuscript.

Kind regards,

Julia A. Jones

Academic Editor

PLOS ONE

4. Please include captions for your all Supporting Information files at the end of your manuscript, and update any in-text citations to match accordingly. Please see our Supporting Information guidelines for more information: http://journals.plos.org/plosone/s/supporting-information.

Reviewers' comments:

Reviewer's Responses to Questions

**Comments to the Author**

1. Is the manuscript technically sound, and do the data support the conclusions?

Reviewer #1: Yes

Reviewer #2: Yes

2. Has the statistical analysis been performed appropriately and rigorously? 

Reviewer #1: Yes

Reviewer #2: Yes

3. Have the authors made all data underlying the findings in their manuscript fully available?

Reviewer #1: No

Reviewer #2: Yes

4. Is the manuscript presented in an intelligible fashion and written in standard English?

Reviewer #1: Yes

Reviewer #2: Yes

5. Review Comments to the Author

Reviewer #1: Authors state data will be made available after review otherwise my comments are included in the attached pdf. This is a very interesting study that I feel will be relevant for the ecoacoustic community.

Reviewer #2: Overall: This paper presents unique work on integrating structured and unstructured sources of avian occurrence data in the Neotropics. The authors find that supplementing acoustic data-based distribution models with citizen science eBird data increases their flexibility with only minor predictive penalties. These conclusions seem well supported by the data and analysis. The methods section is well-detailed (almost too much so), but the results and discussion sections don’t reflect the same level of thoroughness. Careful thought should be given to making the manuscript more targeted and concise; I believe the issue here, for me at least, is less of scientific quality (the work is quite rigorous, novel, and provides a first step in a hopeful direction for neotropical ornithology), and more of clear communication of ideas and acknowledgement of assumptions. Additionally, because this is one of the first times eBird data has been integrated with any type of data resulting from structure surveys in the neotropics, I would suggest the authors improve their discussion of the limitations of neotropical eBird data, which is influenced by different sources of bias than that from north temperate regions. I would be interested in seeing future iterations of this very interesting first draft.

Introduction: This section presents clear background information on the methodologies, identifies a deficiency of this type of work in the neotropics, and states the intentions of the study. The narrative flow is generally good and easy to follow, but I suggest several considerations below.

Lines 59-64: This paragraph seems out of place to me. It interrupts the flow from the identification of increasing data availability and innovation of new tools in the first paragraph to the introduction of ARUs as tool to be used in the tropics. It would make sense to include this type of study region information at the beginning of the last paragraph in the Introduction as a way to further justify your efforts.

Line 67: You say that ARUs are more effective than humans at detecting birds in dense understory conditions but later contradict this idea in the Discussion by acknowledging the benefits of human observers.

Lines 68-71: This is fine information to include (e.g. justification of ARU usage in the tropics), but throughout the remainder of the paper conservation/survey urgency in relation to this work is not mentioned again so I wonder how beneficial it is here.

Lines 76-81: These two sentences duplicate ideas. I’d suggest condensing them into one consistent idea.

Line 92: spell out MODIS

Line 100: “data pooling has been increased the…” change to “data pooling has increased…”

Lines 119-120: While curiosity is an important component of this research, I think rephrasing this thought to something more direct and objective-like would be better for this section.

Lines 123-125: To what extent and where is this done in the manuscript? Maybe I am missing it in the discussion.

Methods: Very detailed and highly repeatable although I feel that some structure is needed for readers to understand the justification and objectives of each successive step in your modeling process. At times, it is overly dense, and the overarching methodology is sometimes muddied with small side details. Some of the finer details may be best suited for the Supplemental Material section. It is also overdeveloped in comparison to other sections, especially the results and discussion. See line-by-line comments below:

Lines 184-186: I suggest the latter half of this sentence to be changed to “… for the three seasons are available in *citation 14* and in S1 File”

Lines 188-196: This paragraph might fit better at the end of the methods section.

Lines 200-204: Add brief description of study area in relation to a larger geographic area as referenced in the figure.

Lines 216-219: I feel this and the following methodological explanation could be more clearly justified in at least one preceding sentences; some readers may be confused by the need for modeling and model selection here.

Lines 239-242: This information should be stated earlier so readers are aware of this sooner.

Lines 243-250: Not major, but this entire paragraph seems like something that could easily be reduced to one sentence. Some information here does not seem critical.

Lines 249-250: “zero-filled” is not referenced again in the manuscript.

Lines 253-260: Why did you filter eBird data using these criteria? Are you basing this off previous work? You discuss this briefly in the Discussion, but more justification here is desired. eBird data from the tropics is notoriously biased by checklists submitted by birders from other regions who are less experienced with the local avifauna, or guides who inflate their lists in an effort to make their services more appealing. How much quality control was done on the eBird data? I think these ideas are worth addressing, if not in the Methods, then with more detail in the Discussion than is currently present. The following paper provides some structure on this process in a north temperate study: Hallman, T.A., Robinson, W.D. Supplemental structured surveys and pre-existing detection models improve fine-scale density and population estimation with opportunistic community science data. Sci Rep 14, 11070 (2024). https://doi.org/10.1038/s41598-024-61582-6

Lines 260-263: What are you defining as present on a checklist? What about checklists with “Xs”?

Lines 264-267: True, this is the case in north temperate eBirding communities where the majority of checklists are submitted by those very familiar with the local species, but can the same assumption be made for the neotropics?

Lines 273-277: This sentence could be condensed.

Line 275: Is “short” the right word choice here?

Line 278: terra firme is italicized here but not consistently elsewhere is the manuscript.

Lines 316-317: Provide a citation for this sentence.

Line 317: “proportional” seems like not the correct word here, maybe “synonymous”?

Lines 345-346: If you are including traveling checklists with < 1km distance, this buffer needs to be expanded to 1 km. When using data from eBird hotspots you should also consider the implications of extracting habitat predictor data at the coordinates of the checklist. This should be reflected here or in more detail in the discussion.

Lines 376-387: this paragraph is out of place in the Methods section. Either cut it or condense these ideas into a sentence or to two to preface the following ideas.

Line 377: maybe spell out percent here.

Line 379: change colon to semi-colon

Lines 399-402: This is not valid justification for your choice. Rather than acknowledging a lack of literature, explain why you selected 90 days and not shorter or longer.

Lines 414-416: I’m uncertain why you made this choice. Could you perhaps provide some clarification here?

Lines 425-431: This information belongs in either the results or Supplemental Material.

Lines 458-460: Do song rates change in the wet and dry seasons? This likely impacts detectability, no?

Results: Mostly well written and clear. It is comparatively sparse given the detail in the methods section. Are the results from all methods reported or at least referenced as Supplemental Material? There also seems to be a significant amount of interpretation which should be limited to the discussion.

Line 481: “respectively” is not needed.

Line 492: the idea in parentheses should be explored in the Discussion and excluded here.

Lines 510-520: This entire paragraph seems better suited for the Discussion.

Lines 510-513: State this idea more directly; “Importance values of predictors indicates that…”

Line 562: Maybe spell out percent here

Line 571: Quantify “good job” here. This is an example of where the results section verges on the side of too much interpretation.

Discussion: This section touches on key ideas mentioned throughout the paper, but I feel the writing and ideas could be improved and tightened. Like the Results, it feels under-developed in comparison to the Methods with assumptions made in that section unexplored and not fully acknowledged.

Lines 599-600: This sentence could be stronger and more concise; “tropical species as a whole” is clunky.

Lines 612-616: This point is not clearly communicated and I am confused by this sentence.

Lines 616-618: This sentence should include a citation.

Line 624: Is “Tropics” intended to indicate Neotropics? If so, please correct.

Lines 599-625: This entire paragraph could be tighter. I get lost in smaller details and lengthy sentences. The focus here should be on punchy, big take-aways.

Lines 632-638: This sentence is much too long.

Line 644: Change “This be evidence…” to “This is evidence…”

Lines 653-661: Again, this sentence is much too long.

Line 664: Spell out TNC, also add mention of USGS Breeding Bird Survey, and other regional highly structured surveys (such as in the cited paper above).

Lines 687-689: Are any of these species known to lek, if so identify them? What about variation in vocalization rates throughout wet-dry seasons? In other parts of the neotropics, the wet season dawn choruses are characteristically reduced in length and intensity.

Lines 692-695: This is interesting but not necessarily useful here.

Lines 696-711: This paragraph introduces too much new information. Yes, it puts this study in the broader conservation context, but this can be done more concisely while also restating the major findings.

6. PLOS authors have the option to publish the peer review history of their article (what does this mean? ). If published, this will include your full peer review and any attached files.

**Do you want your identity to be public for this peer review?** For information about this choice, including consent withdrawal, please see our Privacy Policy .

Reviewer #1: No

Reviewer #2: **Yes: ** Nolan Michael Clements

---

## [Author Response · Author response to Decision Letter 1]

25 Apr 2025

Please see the document titled "Response to Reviewers" in the resubmission for a full rebuttal of reviewer comments.

---

## [Decision Letter · Decision Letter 1]

PONE-D-24-56728R1Combining acoustic survey and citizen science data yields enhanced species distribution models for tropical rainforest birdsPLOS ONE

Dear Dr. Rumelt,

Thank you for submitting your manuscript to PLOS ONE. After careful consideration, we feel that it has merit but does not fully meet PLOS ONE’s publication criteria as it currently stands. Therefore, we invite you to submit a revised version of the manuscript that addresses the points raised during the review process.

The revised version that you submitted contained additional analyses in response to reviewer comments, which generated additional minor revision suggestions.  If these can be addressed, it may not be necessary to send the manuscript out for additional review.

We look forward to receiving your revised manuscript.

Kind regards,

Julia A. Jones

Academic Editor

PLOS ONE

Journal Requirements:

Additional Editor Comments:

The revised manuscript had new analyses in response to reviewer comments, which generated some minor revision suggestions. If these can be addressed, it should not be necessary to send the R2 version out for another review.

Reviewers' comments:

Reviewer's Responses to Questions

**Comments to the Author**

1. If the authors have adequately addressed your comments raised in a previous round of review and you feel that this manuscript is now acceptable for publication, you may indicate that here to bypass the “Comments to the Author” section, enter your conflict of interest statement in the “Confidential to Editor” section, and submit your "Accept" recommendation.

Reviewer #1: (No Response)

Reviewer #2: All comments have been addressed

2. Is the manuscript technically sound, and do the data support the conclusions?

Reviewer #1: Partly

Reviewer #2: Yes

3. Has the statistical analysis been performed appropriately and rigorously? 

Reviewer #1: Yes

Reviewer #2: Yes

4. Have the authors made all data underlying the findings in their manuscript fully available?

Reviewer #1: Yes

Reviewer #2: Yes

5. Is the manuscript presented in an intelligible fashion and written in standard English?

Reviewer #1: (No Response)

Reviewer #2: Yes

6. Review Comments to the Author

Reviewer #1: The authors have spent considerable time and effort on their revisions and responses to questions and I find responses thoroughly addressed my original comments. I appreciate the author’s extension of the modeling to now incorporate multiple spatial summaries of the data as the importance of different buffer sizes can vary. The addition of the 1km analysis strengthens the generalizability of the study; however, it adds significant analysis and results which require minor attention.

I have minor comments related to the new analysis/changes:

- 297: GEDI lidar: GEDI should be spelled out

- 392-423, Table S3, Fig4: Even though RF is extremely robust against multicollinearity, interpretation of variable importance using Gini impurity scores can be highly impacted by multicollinearity, where correlated predictors that are important can “share” importance among the highly correlated predictors resulting in importance dilution (e.g., doi.org/10.1186/1471-2105-11-110). This may make interpretability of the novel 1km predictors less reliable. Table S3 correlations are frequently >0.8 which is highly correlated and may impact variable importance, especially considering some of those variables are frequently important in fig S6. I do not think remodeling necessary, particularly because the discussion focuses more on the findings from the occupancy modeling, but the results section does include the encounter model findings and it is important to acknowledge that variable importance with the addition of the 1km covariates introduces possibly confounding effects on variable importance interpretations. If the authors chose to update the variable importances, the permutation importance option in ranger is more robust to multicollinearity than impurity.

- Fig4: it would be helpful to keep the facet panel titles consistent in the figure and the caption (e.g., in C “Other” in figure but “non-forest habitat predictors” in caption). Also the case in 589-599 in the results which mirrors the caption.

- Table 3: it appears the caption is below the table

Reviewer #2: This is a much-improved draft, with tighter writing and nice adjustments to the modeling. I have no additional comments on the Abstract, Introduction, or Results sections, and only a few limited notes overall.

Line 84: Please spell out MODIS here, rather than only in the Methods section. This is my oversight — I made a similar comment regarding the reference in the Methods but didn’t catch this instance.

The Methods section now feels clearer, and the progression of information is more intuitive. I wonder if a logical workflow diagram in the supplementary materials might help readers follow the many steps involved in data processing, pooling, and analysis. For a proof-of-concept paper like this, I think a visualization of the methods might help others who hope to implement similar tools in the tropics. This is by no means a requirement — just a suggestion if time permits.

Lines 778–784: This is a fantastic point that many eBird users fail to recognize.

Throughout the discussion you do a really nice job of optimistically identifying the shortcomings of using eBird data from the tropics without undercutting your own work. I think part of my first-round grouchiness surrounding the assumptions of eBird data was mostly driven by inherent differences in how citizen observers collect and submit data in temperate versus tropical systems – a process which is not fully acknowledged in the literature. Since you are one of the first to use tropical eBird data for any purpose which involves rigorous models, your thoughts really set a good precedent for what others should expect or account for.

This is really interesting work with clear and impactful results. It has been a pleasure to review this manuscript and watch it improve with each revision.

7. PLOS authors have the option to publish the peer review history of their article (what does this mean? ). If published, this will include your full peer review and any attached files.

**Do you want your identity to be public for this peer review?** For information about this choice, including consent withdrawal, please see our Privacy Policy .

Reviewer #1: No

Reviewer #2: **Yes: ** Nolan M. Clements

---

## [Author Response · Author response to Decision Letter 2]

21 Jun 2025

Responses to Reviewer 1:

The authors have spent considerable time and effort on their revisions and responses to questions and I find responses thoroughly addressed my original comments. I appreciate the author’s extension of the modeling to now incorporate multiple spatial summaries of the data as the importance of different buffer sizes can vary. The addition of the 1km analysis strengthens the generalizability of the study; however, it adds significant analysis and results which require minor attention.

297: GEDI lidar: GEDI should be spelled out

I have replaced the acronym “GEDI” with “Global Ecosystem Dynamics Investigation (GEDI)”

392-423, Table S3, Fig4: Even though RF is extremely robust against multicollinearity, interpretation of variable importance using Gini impurity scores can be highly impacted by multicollinearity, where correlated predictors that are important can “share” importance among the highly correlated predictors resulting in importance dilution (e.g., doi.org/10.1186/1471-2105-11-110). This may make interpretability of the novel 1km predictors less reliable. Table S3 correlations are frequently >0.8 which is highly correlated and may impact variable importance, especially considering some of those variables are frequently important in fig S6. I do not think remodeling necessary, particularly because the discussion focuses more on the findings from the occupancy modeling, but the results section does include the encounter model findings and it is important to acknowledge that variable importance with the addition of the 1km covariates introduces possibly confounding effects on variable importance interpretations. If the authors chose to update the variable importances, the permutation importance option in ranger is more robust to multicollinearity than impurity.

I think this is a very valid point to bring up. I don’t think it would be feasible to include a permutation-based assessment at this stage of the analysis as, being that this calculation is taking place within every bootstrap, doing so could lead to exponential increases to what is already a very long run time. This scared me when I was originally designing the analysis and is ultimately why I chose to use impurity scores instead, but you’re right that I need to offer some explanation for this decision. I have edited the beginning of the paragraph starting on line 392 to include the following disclaimer:

“Finally, we used the “importance” functionality within ‘ranger’ to calculate relative predictor importance within each model. We chose to measure predictor importance using the Gini impurity score implementation, which describes the mean decrease in model accuracy when the chosen predictor is excluded during the fitting process. While the impurity score calculation is highly computationally efficient, it is important to note that Gini impurity scores can suffer from importance dilution when predictors are highly correlated [105]. Our decision to use a bootstrapping procedure necessitated making this tradeoff; however, other researchers seeking to replicate these methods may be better served by using the permutation importance method, which is slower but more robust against multicollinearity [101].”

To reiterate this point, I have also added the following text at the end of the figure caption for Fig 4 :

“For computational efficiency, values represent estimated marginal means of Gini impurity scores across all model bootstraps (n=100); note that importance values may be affected by importance dilution due to multicollinearity.”

Fig4: it would be helpful to keep the facet panel titles consistent in the figure and the caption (e.g., in C “Other” in figure but “non-forest habitat predictors” in caption). Also the case in 589-599 in the results which mirrors the caption.

I have edited Fig4 to make the facet titles consistent with the caption and results section text. I have also changed the latter to call them “FCIs” instead of “FCI habitat predictors” to better reflect the facet text.

Table 3: it appears the caption is below the table

Not sure what happened to cause that, but it’s fixed now.

Responses to Reviewer 2:

This is a much-improved draft, with tighter writing and nice adjustments to the modeling. I have no additional comments on the Abstract, Introduction, or Results sections, and only a few limited notes overall. The Methods section now feels clearer, and the progression of information is more intuitive. I wonder if a logical workflow diagram in the supplementary materials might help readers follow the many steps involved in data processing, pooling, and analysis. For a proof-of-concept paper like this, I think a visualization of the methods might help others who hope to implement similar tools in the tropics. This is by no means a requirement — just a suggestion if time permits. Throughout the discussion you do a really nice job of optimistically identifying the shortcomings of using eBird data from the tropics without undercutting your own work. I think part of my first-round grouchiness surrounding the assumptions of eBird data was mostly driven by inherent differences in how citizen observers collect and submit data in temperate versus tropical systems – a process which is not fully acknowledged in the literature. Since you are one of the first to use tropical eBird data for any purpose which involves rigorous models, your thoughts really set a good precedent for what others should expect or account for. This is really interesting work with clear and impactful results. It has been a pleasure to review this manuscript and watch it improve with each revision.

Line 84: Please spell out MODIS here, rather than only in the Methods section. This is my oversight — I made a similar comment regarding the reference in the Methods but didn’t catch this instance.

No problem, I’ve edited this line so that it is now fully spelled out

Lines 778–784: This is a fantastic point that many eBird users fail to recognize.

Thanks… the eBird team is able to disentangle some of this lack of spatial specificity in their in-house modeling analyses because they can directly access the GPS tracks of checklists submitted through the mobile app. As much as I agree with their decision not to release this information publicly for privacy reasons, I am still envious of the cool things they can do with it.

---

## [Editor Report · Decision Letter 2]

Combining acoustic survey and citizen science data yields enhanced species distribution models for tropical rainforest birds

PONE-D-24-56728R2

Dear Dr. Rumelt,

We’re pleased to inform you that your manuscript has been judged scientifically suitable for publication and will be formally accepted for publication once it meets all outstanding technical requirements.

Kind regards,

Julia A. Jones

Academic Editor

PLOS ONE
---

## [Editor Report · Acceptance letter]

PONE-D-24-56728R2

PLOS ONE

Dear Dr. Rumelt,

I'm pleased to inform you that your manuscript has been deemed suitable for publication in PLOS ONE. Congratulations! Your manuscript is now being handed over to our production team.

Kind regards,

on behalf of

Dr. Julia A. Jones

Academic Editor

PLOS ONE